



# Mesoscale permeability variations estimated from natural airflows in the decorated Cosquer Cave (SE France)

Hugo Pellet [1, 2], Bruno Arfib [1], Pierre Henry [1], Stéphanie Touron [2], Ghislain Gassier [1]

[1] Aix Marseille Univ, CNRS, IRD, INRAE, CEREGE, Aix-en-Provence, France

[2] UAR3224-CRC-Laboratoire de Recherche des Monuments Historiques, France

*Correspondence to*: Hugo Pellet (pellet@cerege.fr)

**Abstract.** Conservation of decorated caves is highly dependent on airflows in the karst network and through the surrounding host rock. Airflows are driven by pressure gradient and influenced by the shape of the karst conduits and the permeability of the carbonate rock massif. The Cosquer cave is an Upper Paleolithic decorated cave, half drowned in a coastal karst, where

conservation is also dependent on the cave's pools connected to the sea. Hydroclimatic data, such as air pressure and temperature and water level inside and outside the cave have been measured for several years to identify the main processes governing the water level variations, the airflows and the air renewal. Data show an unusual behavior for a karst: the karst air pressure is nearly always higher than the atmospheric pressure. As a result, the water level in the cave is below the sea level. The daily variations of the sea tide provide an assessment of the cave volume above the pools water level. Although the cave

air is confined by the rock and the seawater, there are also external air inflows during short pressurization events. Moreover, the carbonate rocks effective permeability to air at the massif scale is inferred from the cave air pressure decrease over the summer season, by applying Darcy's law in a partially-saturated medium. Six years of data show that permeability varies from year to year, and according to the cumulated rainfalls during the spring and summer. The driest years are correlated with a higher permeability, a faster air pressure decrease in the cave and a faster rise of the pools water level. In the future, in

the context of climate change, a perturbation of the rock permeability is then expected in the near surface caves, which will impact airflows in decorated caves and may alter their fragile hydroclimatic stability.

## 1 Introduction

Upper Paleolithic decorated caves constitute an exceptional cultural heritage. The climatic stability of the subterranean karstic environment has ensured the well-preservation of paintings and engravings for millennia, despite their vulnerability

(Andrieux, 1977; Mangin and Andrieux, 1984; Baffier, 2005; Bourges et al., 2006b). The equilibrium of the cave's climate is delicate and maintained through intricate interactions with the environment (Quindos et al., 1987; Bourges et al., 2006a, 2014; Peyraube et al., 2018; Leplat et al., 2019). Disruptions in the climatic equilibrium were notably linked to cave equipment and tourism (Cigna, 1993; Baker and Genty, 1998; Touron et al., 2019) yet it does not entirely spare caves closed





to the public, especially in a context of climate change (Domínguez-Villar et al., 2015; Bourges and Enjalbert, 2020a). A

shift in the climatic equilibrium could lead to fluctuations of different parameters such as $CO_2$ concentrations, humidity, temperature or air exchanges and flows (Badino, 2010; Mattey et al., 2013; Kukuljan et al., 2021). This, in turn, could contribute to the deterioration of the artwork due to outbreaks of microorganisms (Lefèvre, 1974; Martin-Sanchez et al., 2012; Borderie et al., 2015), the emergence of efflorescence (Lepinay et al., 2018; Germinario and Oguchi, 2021) or the processes of calcite precipitation or dissolution (Ford and Williams, 2007; Touron and Frouin, 2022).

In karst, two main types of airflows occur: airflows between the outside and inside of caves (and conversely) and airflows within the caves (Lismonde, 2002; Sainz et al., 2018; Gázquez et al., 2022). In caves with 2 openings, the airflow is mainly driven by the temperature difference between the outside and inside of the cave or between the upper and lower openings (Lismonde, 2002; Gabrovšek, 2023). The flow is continuous and its direction depends on the season. Within caves with single-aperture, airflows result from air density gradient due to difference in temperature or humidity either within the cave

or between the cave and the outside environment (Lismonde, 2002; Luetscher and Jeannin, 2004; Malaurent et al., 2006; Liñán et al., 2018; Huang, 2018). These flows are subject to seasonality, with generally stronger flows in winter and stratification of air masses in summer (Perrier et al., 2007; Mattey et al., 2013; Lacanette et al., 2023). These air movements also occur in caves artificially closed but are comparatively milder in intensity.

Exchanges with the outside environment resulting in the air renewal of the caves, as well as exchanges between the different

rooms, are conventionally assessed through measurements of radon and/or $CO_2$ concentrations (Richon et al., 2005; Kowalczk and Froelich, 2010; Sainz et al., 2018). Air renewal can be calculated when the cave volumes are known, but these latter are not systematically measured, as this can be time-consuming and costly. Cave volumes can be obtained using different techniques such as lasergrammetry, photogrammetry (Mohammed Oludare and Pradhan, 2016) or estimated from the 3D speleological hand-survey, but they stay limited by accessibility for human investigation. The permeability of the

host rock also influences exchanges with the outside environment and can be locally measured, at the scale of a few centimeters on plugs (Borgomano et al., 2013), or at the scale of a well using pumping tests in the saturated zone or in the unsaturated zone (Kuang et al. 2013). Permeability to air or to water of the unsaturated zone is also dependent on the water content. Thus, dealing with conservation of decorated caves, permeability of the carbonate massif in the unsaturated zone is a key parameter since it can control the air or water flows through the rock, by limiting or enhancing the exchanges flux.

This paper aims to estimate the volume of a coastal cave, determine the net airflow exchanged with the outside environment and discuss the variation of the effective air permeability of the massif. The methodology relies on the monitoring of in situ pressure and temperature data.

The topic is explored by studying the Cosquer cave, which is a singular case of a decorated cave located within a partially submerged coastal karst. Although other partially submerged caves have been found in the Mediterranean region (Arfib and

Charlier, 2016; Castagnino Berlinghieri et al., 2020; Arfib and Mocochain, 2022), the Cosquer cave is of special interest since it is isolated by siphons on one side and by low-permeability limestone massif on the other side. First data (Vouvé et al., 1996; Arfib et al., 2018) have shown that the air pressure in the cave can remain higher than the outside atmospheric



pressure for weeks. They also highlighted that outside air rapidly flows into the cave during brief events, resulting in an increase of cave air pressure. Since the massif is not airtight, air slowly flows out through the limestone massif. By taking advantage of this particular behavior, we can thus investigate the permeability variations of a karst massif at the mesoscale. Firstly, we present two full years of data including cave air pressure, atmospheric pressure, cave water level variations and sea level variations. These data are used to investigate cave pressure fluctuations across annual to daily temporal scales. The daily pressure variations related to tides provide an assessment of the cave volume filled by pressurized air above the water level of the pools, applying the ideal gas law. This outcome is used to compute the net airflows entering and leaving the

cave. Finally, the limestone effective air permeability at the massif scale is estimated from the cave air pressure decrease during the summer season, applying the Darcy's law. Permeability is then compared to rainfalls and its evolution is discussed in the context of climate change. This study gives for the first time a conceptual model and a quantitative assessment of flows within both the saturated and unsaturated zones of the Cosquer cave. It also highlights that airflows may change in karst unsaturated zone with changes in the water cycle.

**2 Study site and data**

The Cosquer cave, located in the South-East of France in the Mediterranean seashore, is a partly submerged cave. It is located in the Calanques National Park, nearby Marseille city in Provence. Part of the cave develops below the current sea level and is filled by seawater, and part of the cave still remains filled with air above the sea level. It hosts a large range of paintings and engravings from the Upper Paleolithic with 553 graphic entities recorded to date, dating back 32 500 to 19 000

years B.P. (Clottes et al., 1992a, 1992b, 1997; Valladas et al., 2001, 2017). Among representations, some animals rarely represented in prehistorical artwork (penguins and jellyfish) are painted (Clottes et al., 1992a; Delporte et al., 1994). All the preserved artworks are located in the aerial part of the cave. The only way to currently access the cave in the limestone massif is by cave diving in submerged karstic conduits. However, during the Upper Paleolithic period, prehistoric humans accessed the cave by an entrance that is now 37 meters below the sea level. The sea level was back then lower, e.g. during

the Last Glacial Maximum about 20 000 years ago and about 120 meters lower than today (Benjamin et al., 2017). The entrance has been flooded between 10 000 and 8 000 years BP (Sartoretto et al., 1995; Lambeck and Bard, 2000) by rising sea level.





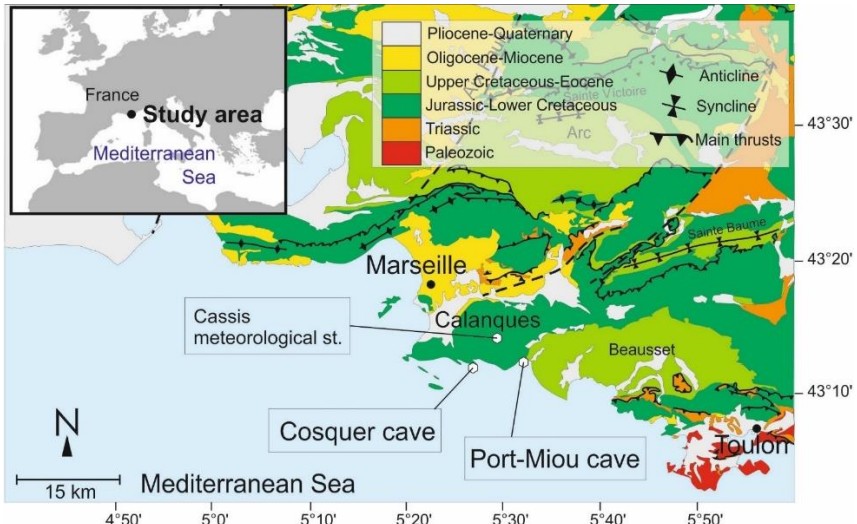

**Figure 1: Structural map of the Provence region (south east of France), with location of the Cosquer cave, the Port-Miou**
**observatory and the Cassis meteorological station. Modified from Lamarche et al., 2012.**

The cave is embedded in the Morgiou massif, a peninsula made of early cretaceous urgonian limestones (Masse et al., 2020).
These limestones are tight carbonates, rudist-rich oolitic grainstones. According to thin-sections observations these
carbonates do not display porosity (Lamarche et al., 2012; Matonti et al., 2015). The cave consists of two rooms whose walls
host paleolithic paintings and engravings and pools hydraulically connected to the sea. Karstic voids used bedding planes
and fractures to develop, forming a karst network made of four areas: (1) a main conduit from 37 meters deep to the Room 1,
below the sea level, (2) Rooms 1 and 2, partly flooded, containing paleolithic decorations above the pools water level, (3) a
vertical shaft (named "Grand Puits", or "high shaft" crossing the Room 2) that is 35 meters high above the pool water level,
(4) an upper karst conduit located in a higher bedding plane, running above the decorated rooms and connected on one side
to Room 2 by the Grand Puits and to a small pool of seawater on the other side.




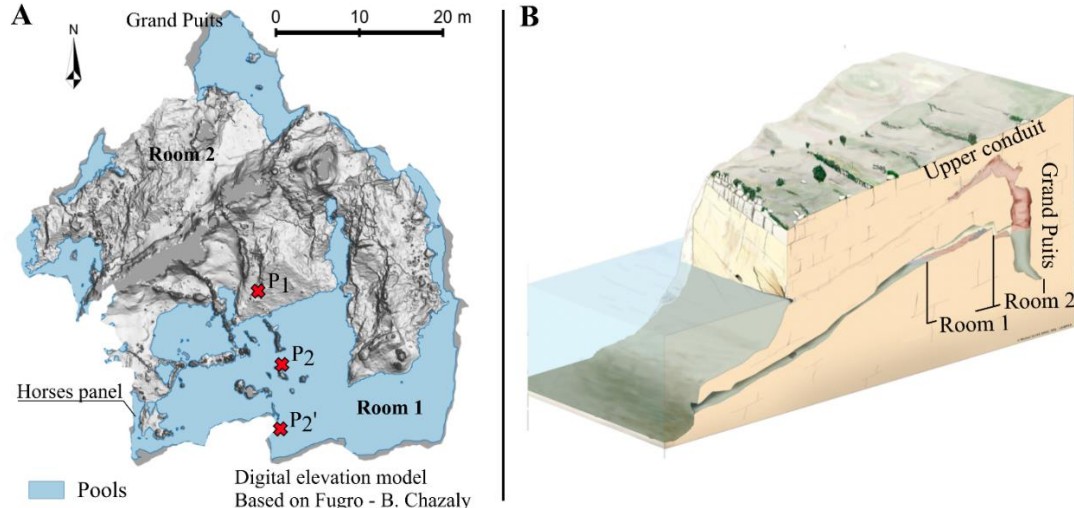

**Figure 2: (A) Topographic map of the Cosquer cave. Blue areas are the pools. Probes in air are located in P₁, hooked to a stalagmite 5 m away from the water and 1 m above the ground. The probes in water are located in P₂ and P₂' (map designed by C. Font, 2023, Équipe grotte Cosquer). (B) Schematic cross-section of the Cosquer cave. The entrance of the conduit is 37 m below the sea level. Modified from Olive and Vanrell, 2021.**

Temperature and pressure in air are measured in the Cosquer cave using respectively a Cera-Diver probe (resolution of 0.01 °C and an accuracy of ± 0.2 °C) and a STS DLN probe (resolution of 10 Pa and an accuracy of +/- 0.1 % of full-scale (+/- 130 Pa)). Both probes are located in Room 1, hooked to a stalagmite about 5 m away from the pool, 1 m above the ground and 10 cm away from the stalagmite (location P₁ on Fig. 2A). CTD-Diver probes measure absolute pressure, temperature and specific electric conductivity of water in Room 1 (locations P₂ and P₂' on Fig. 2A). CTD pressure resolution

is 10 Pa and accuracy 50 Pa. Atmospheric air pressure outside the Cosquer cave (SNO KARST, 2021) and sea level are measured 5 km away from the cave using Cera-Diver or TD-Diver probes, in the Port-Miou observatory of the French Karst National Observatory Service (Fig. 1). Port-Miou sea probes are moored in a large karst conduit connected to the sea (Arfib and Charlier, 2016; Jourde et al., 2018). This location protects measurement from marine storms and waves. Data are continuously recorded since 2014 at a 5-minutes time-step. Height of water column are calculated from absolute pressure by

substracting the air pressure above the water table and converted to meters of sea water (m$_{sw}$) using density of the Mediterranean Sea ($\rho_{sea}$ = 1027 kg m$^{-3}$). Probes are factory calibrated and their clocks synchronized. Measurement and clock drifts are checked with control probe during data collection (every 4 to 5 months). Available data run from 2014 to 2020 for this study. Precipitation data is provided by Météo-France records at Cassis city (Météo-France, 2023) located 5 km to the North-East of the Cosquer cave in the Calanques massif.

All the parameters used in this paper are summarized in Table 1 and illustrated on Fig. 3.





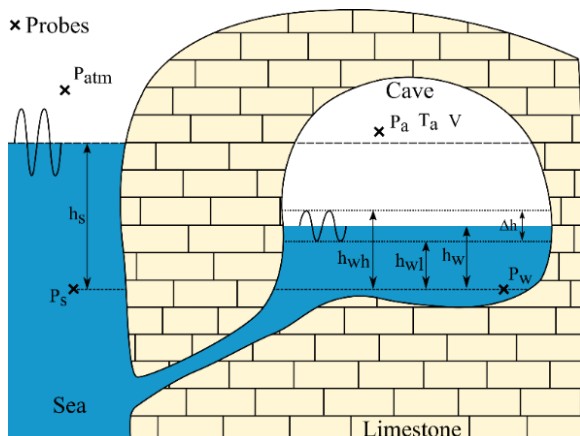

**Figure 3: Main parameters measured with pressure probes, viewed on a conceptual cross section of the coastal cave. Referred to Table 1 for parameters explanation.**






| Symbol | Parameters | Unit |
|---|---|---|
| $P_{atm}$ | Atmospheric pressure (outside the cave) | Pa or $m_{sw}$ |
| $P_s$ | Absolute pressure above the probe moored in the sea | Pa or $m_{sw}$ |
| $h_s$ | Sea level above the probe | $m_{sw}$ |
| $P_a$ | Cave air pressure | Pa or $m_{sw}$ |
| $P_w$ | Cave absolute pressure above the probe moored in water | Pa or $m_{sw}$ |
| $h_w$ | Cave water level above the probe | $m_{sw}$ |
| $h_{wl}$ | Cave water level at low tide | $m_{sw}$ |
| $h_{wh}$ | Cave water level at high tide | $m_{sw}$ |
| $\Delta h_w$ | Tide range in cave | $m_{sw}$ |
| $T_a$ | Cave air temperature | °C |
| $V_l$ | Cave volume at low tide | $m^3$ |
| $V_h$ | Cave volume at high tide | $m^3$ |
| $V$ | Cave volume | $m^3$ |
| $h_0$ | Reference water level above the probe | $m_{sw}$ |
| $V_0$ | Reference cave volume | $m^3$ |
| $S_w$ | Surface of water bodies in the cave | $m^2$ |
| $n$ | Cave air quantity | mol |
| $Q$ | Volumetric airflow rate | $m^3\ s^{-1}$ |
| $Q_n$ | Molar flow rate | $mol\ s^{-1}$ |
| $q_n$ | Molar flux | $mol\ m^{-2}\ s^{-1}$ |
| $L$ | Fracture length or limestone thickness | m |
| $W$ | Fracture width | m |
| $A$ | Cave cross-sectional area | $m^2$ |
| $\lambda_a$ | Air intrinsic transmissivity | $m^3$ |
| $k_a$ | Air effective permeability | $m^2$ |
| $b$ | Hydraulic aperture of a fracture | m |
| $r$ | Radius of a pipe (equivalent to a karst conduit) | m |
| $\mu$ | Air dynamic viscosity | Pa.s |
| $R$ | Ideal gas constant | $J.K^{-1}.mol^{-1}$ |
| $\rho_{sea}$ | Sea water density | $kg.m^{-3}$ |
| $g$ | Gravitational acceleration | $m^2.s^{-1}$ |
| $\gamma$ | Adiabatic index | - |
| $P_0$ | Standard pressure | Pa |
| $T_0$ | Standard temperature | K |

**Table 1: Presentation of the physical parameters, their notation and units used in this paper.**





## 3 Cosquer cave hydroclimate

Figure 4 shows pressure, temperature and water level time series recorded in 2017 and 2018. These two years are used to illustrate the hydroclimatic behavior of the cave. Three types of variations are identified and described below: (1) seasonal
variations, (2) events lasting several hours to several days, (3) daily variations.

### 3.1 Seasonal pressure variations

Data show that air pressure in the Cosquer cave is always higher than outside atmospheric pressure (Fig. 4). This very peculiar feature had already been shown by previous works (Vouvé et al., 1996; Arfib et al., 2018) and has now been confirmed on the timescale of several years of continuous survey (2014-2020). Air pressure in the cave and water level of
the pools are correlated. Between late spring and early autumn, there is a slow decrease in cave air pressure and water level simultaneously increases.

Summer depressurization rate was in average -0.21 $cm_{sw}$ $day^{-1}$ in 2017 and -0.32 $cm_{sw}$ $day^{-1}$ in 2018 (mean over July and August). At the end of summer, cave air pressure is minimal and close to atmospheric pressure outside the cave. A succession of pressure peaks occurs between October and May (highlighted in grey in Fig. 4) and these are generally absent
in summer. These sharp rises in air pressure over tens of minutes to few hours followed by a rapid pressure decay (over a day or so) are referred in this paper as pressurization events and will be described in more detail in the next section. Between 20/10/2017 and 30/04/2018, about 30 of these pressurization events occurred. Net air pressure usually increases in the cave during these events, i.e. the air pressure is usually higher after the event than before. Two thresholds are graphically identified in Fig. 4: (i) maximum air pressure never exceeds 11.5 $m_{sw}$ (1.16 hPa), (ii) immediately after pressurization peaks,
the air pressure drops down to an overpressure level between 10.8 $m_{sw}$ (1.09 hPa) and 10.7 $m_{sw}$ (1.08 hPa). Below this level, the pressure decrease rate slows down considerably. The lowest water level is about 1.5 $m_{sw}$ below the seawater level (0.40 $m_{sw}$ above the probe) during winter. Thus, at a seasonal time scale, the pressure variation range is around 1.5 $m_{sw}$ (0.15 hPa).

Air temperature varies in the range 16°C to 21°C, in a seasonal pattern. The maximum is observed at the end of the summer
and the minimum at the beginning of the spring.



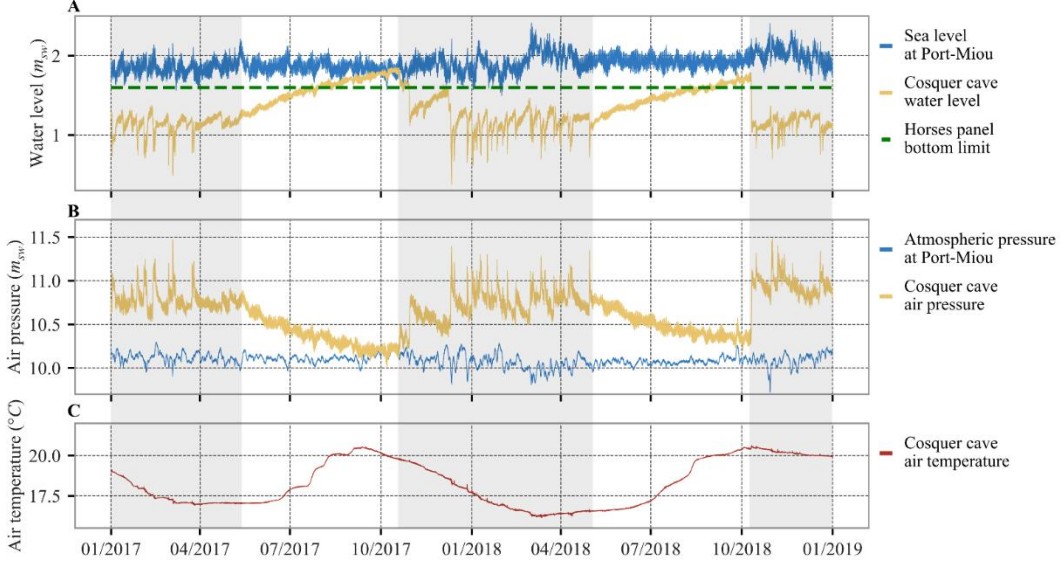

**Figure 4: Pressure, water level and temperature time series recorded in the Cosquer cave and at the Port-Miou observatory for years 2017 and 2018: (A) Sea level at Port-Miou ($h_s$) and Cosquer cave water level ($h_w$), expressed in column of seawater ($m_{sw}$) above the probe with the same reference level. The green dashed line shows the bottom of the horses panel (paleolithic decorated wall). (B) Atmospheric pressure ($P_{atm}$) outside the cave and cave air pressure ($P_a$) (C) cave air temperature ($T_a$). Pressurization events periods are highlighted in grey.**

## 3.2 Pressurization events

Pressurization events can be separated in different stages with analogy to the flood hydrograph (Chow et al., 2013). One example is detailed in this paper to illustrate the phenomenon. This example, lasting 38.3 hours, occurred from April 29 to May 1, 2018. This is a representative pressurization event, with three main stages (Fig. 5) identified by the slope variations of the cave air pressure. The first stage is the pressurization stage corresponding to the rising limb of the curve (from A to D), it lasted 7.8 hours and induced an increase of 73.7 $cm_{sw}$ of air pressure in the cave (mean pressurization rate around 9.4 $cm_{sw}$ $h^{-1}$). The maximum slope of the rising limb reached a maximum pressurization rate close to 15.8 $cm_{sw}$ $h^{-1}$ (Fig. 5, from B to C). After the pressure peak at point D, the second stage started until an inflection point in the recession curve; the pressure dropped rapidly by 57.4 $cm_{sw}$ from D to E (Fig. 5), at an average rate of -1.9 $cm_{sw}$ $h^{-1}$, which is more than 100 times higher than the slow pressure decrease rate described in the previous section (summer time). The third stage started from point E, around 10.8 $m_{sw}$ with a lower pressure decrease rate of -0.2 $cm_{sw}$ $h^{-1}$ and was interrupted by the next pressurization event.





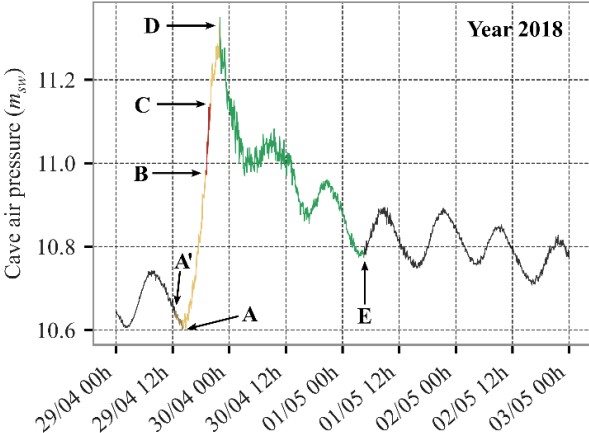

**Figure 5: Example of the pressurization event occurring between 29 April 2018 to 1 May 2018. (A) start of the pressurization event based on cave air pressure variations; (B to C) maximum slope of the rising limb; (D) pressure peak, end of the pressurization stage (A to C) and start of the rapid pressure drop stage (or rapid falling limb), (E) end of the rapid pressure drop stage and start of the slow pressure decrease stage. (A') start of the pressurization event based on cave air quantity computation.**

### 3.3 Variations at tidal scale

Tide induced pressure variations are recorded in the cave. Figure 6 shows in situ pressures expressed in meter of seawater ($m_{sw}$) and mean-centered at a two days scale in order to focus on the relationships between sea level ($h_s$, measured at Port-Miou), cave air pressure ($P_a$), cave absolute pressure in water ($P_w$) and cave water level ($h_w$, computed from $P_a$ and $P_w$). The different parameters are considered in summer when the cave air pressure is low and the cave water level is high (from 12/08/2017 to 14/08/2017, Fig. 6A) and in winter when the cave air pressure is high and the cave water level is low (from 23/12/2017 to 25/12/2017, Fig. 6B). The sea tide is transmitted through the submarine karst conduits or open fractures and bedding planes so the absolute pressure in water inside the Cosquer cave equilibrates with the sea level variations, without any noticeable lag (data time-step is 5 minutes).

The water level in the cave varies less than the tide outside because a part of the pressure variation is transmitted to the confined air above the pools surface. Focusing on the examples in Fig. 6, during summer the mean tide amplitude is 14.1 cm$_{sw}$ outside and 5.6 cm$_{sw}$ inside the cave. During winter the mean tide amplitude is 12.5 cm$_{sw}$ outside the cave and 5.2 cm$_{sw}$ inside. On both seasons, the tide amplitude in the cave is thus about 40 % of the sea tide and variations of the cave air pressure account for the remaining 60%. The magnitude of the damping of the tide in cave depends on the volume of air trapped in the cave.

Tide-related temperature variations are observed. These variations are small, less than 0.05 °C crest to crest (not shown in Fig. 4 or Fig. 6) and display a $\pi/2$ phase advance (3 hours) with respect to tide pressure variations. The amplitude of these variations is much smaller (about twenty times) than the adiabatic temperature variation (Eq. (7)) that would result from the tide air pressure changes. Moreover, simultaneous measurements obtained with 2 probes set 5 cm and 21 cm from the





surface of nearest wall (a stalagmite) recorded the same small temperature variations within <0.01 °C. These observations indicate that thermal convection is very active at least in the decorated rooms of the cave and that at the tide time scale, the

air in the cave remains close to thermal equilibrium with the walls.

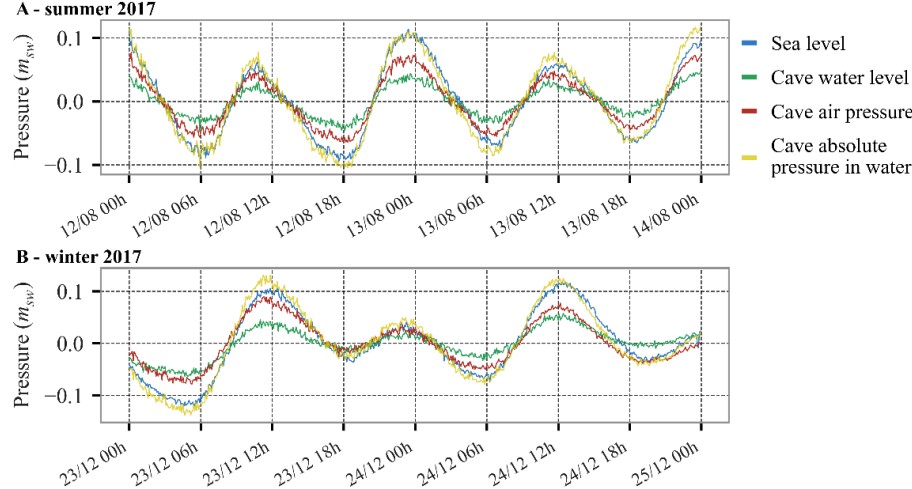

**Figure 6: Time series of the sea level ($h_s$), the cave air pressure ($P_a$), the cave water level ($h_w$) and the absolute pressure in water in the cave ($P_w$) centered by their mean values during (A) a summer period when $P_a$ is low (12/08/2017 to 14/08/2017) and (B) a winter period when $P_a$ is high (23/12/2017 to 25/12/2017).**

## 3.4  A threatened treasure: focus on the Horses panel

We showed that the cave water level in the Cosquer cave is lower than the sea. This behavior limits the submersion of art nearby water such as the horses panel (Fig. 2A). Figure 7 shows, for years 2017 and 2018, the cumulated time (in % of the year) of the water level above the probe (in $m_{sw}$). A scaled photo of the Horses panel is added to the figure. The green dashed line marks the bottom of the painting. Usually, water level in coastal karsts is equal to the sea level; thus in this theoretical

case, the lower part of the horses panel would be continuously flooded. But in the case of the Cosquer cave, air overpressure maintained the horses panel totally out of water 75 % of the time in 2017 and 88 % of the time in 2018 (Fig. 7). The Horses panel is partially flooded from mid-summer, when the cave water level slowly rises up, to the first pressurization event at the end of summer or beginning of autumn (water level above the green dashed line on Fig. 4).

The Horses panel was flooded during 92 days in 2017 and 64 days in 2018. During these periods, the panel undergoes (1)

washout due to the infiltration of water into the rock porosity and (2) mechanical erosion under the effect of alternating wet/dry periods caused by the tide. Currently, air overpressure and tide dumping inside the cave reduce the duration of flooding. The Horses panel is not the only art threatened by water: the Negative Hands panel and archaeological artefacts on the floor are also endangered. The behavior of the karst limits the water level rising on the paintings and engravings, which are still under threat. Global sea level rise is a direct threat to the integrity of the cave.



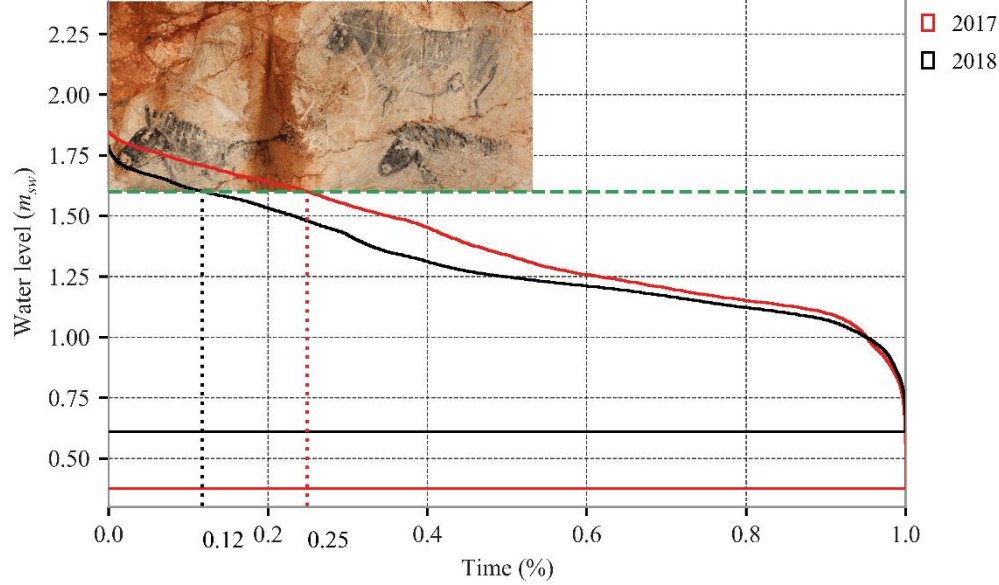

**Figure 7: Cumulated time (in % of the year) of the water level above the probe (in m$_{sw}$) for years 2017 (red) and 2018 (black). The green dashed line marks the bottom of the Horses panel represented on the figure.**

## 4 Model

Recorded data show that air pressure variations in the Cosquer cave are related to the cave volume and the inflow and outflow of air into the confined cave. We aim at calculating the cave volume using the tide variations and the rock permeability using the slow air depressurization stage. This section gives the methods and equations used.

### 4.1 Equations for air pressure and air quantity in the cave

Pressure variations of air in the Cosquer cave are related to variations of air quantity in the cave, temperatures and tides variations . Approximating air as an ideal gas:

$$P_a V = nRT_a \, , \tag{1}$$

where $P_a$ is cave air pressure and $T_a$ is cave air temperature, $n$ is the number of gas moles in the cave, $V$ the volume of the cave, defined as the volume of all the connected voids above the water level, and $R$ is the ideal gas constant. $P_a$ and $T_a$ here represent the average pressure and temperature in the volume of the cave. There is here no requirement with this formulation to assume that $P$ and $T$ are uniform, but the question whether the measurements do represent the average temperature may be asked. This point is dealt with in §4.2. Variations in $n$ correspond to variations of gas content in the cave regardless of the





processes considered. These include air inflows and outflows but potentially also exchanges with the liquid phase by diffusion of dissolved gasses and water liquid/vapor phase change. The effect of water evaporation and condensation may be approximated considering that the air in the cave is generally close to dew point. Using Tetens equation (Monteith and Unsworth, 2013) to calculate water vapor pressure, a maximum range of vapor pressure variations of 18.2 hPa to 24.9 hPa is

obtained for a temperature varying from 16 °C to 21 °C, representing the range of seasonal variations (Fig. 4). This pressure variation corresponds to a 6.7 cm water level change. Conversely, the effect of the thermal expansion of dry air from 16 °C to 21 °C leads to a pressure increase of 17.3 hPa, with ideal gas approximation. At the tidal scale, cave air temperature variations do not exceed 0.05°C and thus changes of water vapor pressure may be neglected. Mean cave air pressure variation because of tides is about 9.5 cm$_{sw}$ (Fig. 6) whereas during slow depressurization (with a depressurization rate of -

0.32 cm$_{sw}$/day) cave air pressure decreases by 0.08 cm$_{sw}$ between 2 tides (~6 h), i.e. 1.4 % of the mean tide range. This variation are therefore neglected.

The total volume of the cave and the number of moles are both unknown. Nevertheless, the variation of volume of the cave due to tidal variations may be estimated from the variations of water height in the cave, which are measured. Knowing $P_a$ and $T_a$ variations, it is thus possible to estimate $n$ assuming it remains constant during a tidal cycle, and hence the total

volume of the cave. On the longer time scale, once this volume is known, the variation of $n$ during the slow depressurization may be calculated from the long-term variations of $P_a$ and $T_a$.

**4.2 Volume calculation**

The variation of the cave volume (filled with air) between high and low tide can be expressed as:

$$V_h = V_l - \int_{h_{wl}}^{h_{wh}} S_w(h) \, dh_w \, , \tag{2}$$

Where $V_h$ and $V_l$ are respectively the cave volume at high and low tide. $h_{wh}$ and $h_{wl}$ are the cave water level at high and low tide. $S_w(h)$ (m²) is the total surface of water bodies (pools) connected to the sea and thus affected by tides. The surface of the pools for a middle stand water level was calculated using QGIS tools from a georeferenced map of the cave and equals to 847 m². For a first order approximation, the variations of pools surface between high and low tide may be neglected because the roof and shore in Room 1, which hosts the largest pool area, are almost parallel (both follow the dip of sedimentary

layers), and the walls of the Grand Puits are subvertical. $S_w(h)$ is thus taken constant:

$$S_w(h) \approx S_w = 847 \, m^2$$

and

$$V_h = V_l - S_w.(h_{wh} - h_{wl}) = V_l - S_w.\Delta h_w \, , \tag{3}$$

Also, $n$ is assumed constant on the time scale of a tide. The validity of this assumption was assessed in section 4.1. Perfect

gas law is applied at high and low tide:

$$\frac{P_{ah}V_h}{T_{ah}} = \frac{P_{al}V_l}{T_{al}} \, , \tag{4}$$





Combining Eq. (3) and Eq. (4) the volume of the cave at low tide $V_l$ is:

$$V_l = S_w \, \Delta h_w \frac{P_{ah}T_{al}}{P_{ah}T_{al}-P_{al}T_{ah}} \, , \tag{5}$$

Except during the transient pressurization events, air pressure within the connected rooms of the cave is at equilibrium.
However, air temperature may not be uniform as it depends on thermal convection for homogenization. To bound volumetric
estimations, we will also consider two end member cases corresponding to isothermal and adiabatic assumptions.

In the isothermal case, Eq. (5) simplifies as:

$$V_l = S_w \, \Delta h_w \frac{1}{1-\frac{P_{al}}{P_{ah}}} \, , \tag{6}$$

In the case of an adiabatic process, there is no heat transfer between the air and the cave walls or water pools and the
variation of temperature of an ideal gas is related to the pressure variation by:

$$\frac{T_{al}}{T_{ah}} = \left(\frac{P_{al}}{P_{ah}}\right)^{1-1/\gamma} \, , \tag{7}$$

The adiabatic coefficient for air at 20 °C is $\gamma = 1.4$ (Lange and Forker, 1967). Combining Eq. (3) and Eq. (7) yields to:

$$V_l = S_w \, \Delta h_w \frac{1}{1-\left(\frac{P_{al}}{P_{ah}}\right)^{1/\gamma}} \, , \tag{8}$$

As air pressure at low tide is lower than at high tide, $P_{al}/P_{ah}$ ratio is less than 1 and the volume estimated with the adiabatic
assumption is larger than for the isothermal one.

### 4.3 Airflow rate

Once the volume of the cave has been estimated from the tidal variations, the quantity of air in the cave is computed over
time as:

$$n(t) = \frac{P(t)}{RT(t)}\left[V_0 + S_w \cdot \left(h_0 - h_w(t)\right)\right] \, , \tag{9}$$

Where $V_0$ is a reference volume of the cave for a reference water level above the probe $h_0$. The net volumetric flow rate $Q$
(m$^3$ s$^{-1}$) into the Cosquer cave (inflow positive) is calculated as:

$$Q = v_m \, Q_n = \frac{RT}{P} Q_n \, , \tag{10}$$

where $v_m$ is the air molar volume (m$^3$ mol$^{-1}$), $Q_n$ the molar flow rate (mol s$^{-1}$) and $R = 8.314$ J K$^{-1}$ mol$^{-1}$ the ideal gas constant.
The net volumetric flow rate can be expressed for standard pressure $P = P_0 = 101325$ Pa and temperature $T = T_0 = 288.15$ K.

### 4.4 Rock mass permeability

The seasonal slow depressurization of the confined Cosquer cave during spring and summer implies air outflows through the
host-rock. The effect of gas compressibility on flow is taken into account approximating air as a perfect gas. Assuming that





airflow follows Darcy's law, and neglecting the hydrostatic gradient in the atmosphere (about 11 Pa m⁻¹), the molar flux may
then be written as (e.g.: Massman, 1989) :

$\quad \boldsymbol{q_n} = \left( \frac{P}{RT} \frac{k_a}{\mu(T)} \right) \mathbf{grad}(P) = \left( \frac{k_a}{\mu(T)RT} \right) \mathbf{grad}\left( \frac{P^2}{2} \right),$ (11)

Where $\boldsymbol{q_n}$ is the molar flux (mol m⁻² s⁻¹), $\mu$ is the dynamic viscosity of air (Pa s), $k_a$ is the medium effective permeability to
air (m²), $T$ the gas temperature (K) and $P$ the air pressure (Pa) . According to the kinetic theory of gasses, the viscosity of a
perfect gas is a function of temperature only and does not depend on pressure (Chapman and Cowling, 1970).

Airflows through a porous medium follow Darcy's law if the pores have a sufficiently low water saturation to host a
continuous gas phase, defining the percolation threshold. The generalization of Darcy's law for air flow in unsaturated
porous medium uses the effective permeability to air ($k_a$):

$k_a = k_{ra} \, k_w \,,$ (12)

$k_{ra}$ is the relative permeability to air (-), describing the influence of air and water content on permeability. It ranges between
0 (at the percolation threshold) to 1 (dry state). $k_w$ is the intrinsic permeability of the host-rock (m²), which is independent of
the fluid properties and saturation. Our data can constrain $k_a$ but not $k_{ra}$. The maximum value of effective permeability to
air from our calculations is thus a lower bound for the intrinsic permeability.

Now considering a steady-state, or slowly varying, flow between a cavity at pressure $P_a$ and the surface at atmospheric
pressure $P_{atm}$, other parameters being held constant ($T$, $\mu$), it follows from Eq. (11) that the total molar flux depends linearly
on the difference between the boundary conditions of the squared pressure. Hence, the effective air transmissivity $\lambda_a$ between
the cave and the ground surface may be defined as:

$\lambda_a = \frac{2\mu RT}{(P_a^2 - P_{atm}^2)} Q_n \,,$ (13)

The variations of temperature in the host-rock are unknown, but lower in amplitude than temperature variations in the cave
and at the ground surface (Bourges et al., 2006a). $T$ is thus taken constant equal to the yearly mean air temperature in the
cave, about 18 °C. In this case $\mu(T) = \mu = 1.81 \ 10^{-5}$ Pa s. This air effective transmissivity coefficient $\lambda_a$ has m³ dimension and
is presented for three flow geometries in this paper.

If pressure gradient is applied over length $L$ (m) on a cross-sectional area $A$ (m²) the air effective permeability (m²) may be
defined as:

$k_a = \lambda_a \frac{L}{A} \,,$ (14)

In the case of the Cosquer cave, $L$ and $A$ reflect the dimensions of the boundary conditions and are illustrated in Fig. 8A. One
possibility is to consider the air effective permeability of the rock volume between the main cave and the ground surface,
hence $A = 2000$ m² and $L = 40$ m. However, it is possible that most of the flow occurs through the upper part of the cave,
closer to the ground and likely less water saturated, namely the top part of the Grand Puits (*high shaft*), that is about 100 m²
and 10 m below the surface according to available 3D models of the cave. The geometric factor $A/L$ may thus be considered
to range from 10 to 50 m.





If a leakage occurring through a fracture of length $L$ and width $W$ as shown in Fig. 8B is considered, the hydraulic aperture of the fracture is (Zimmerman and Bodvarsson, 1996):

$$b = \left(12\lambda_a \frac{L}{W}\right)^{\frac{1}{3}},\tag{15}$$

If a population of N fractures is considered, $b^3$ represents the sum of the cubed apertures of individual fractures. It follows that the larger fractures generally dominate the flow. For instance, if 8 identical fractures of hydraulic opening $b$ are present,

they are equivalent to a single fracture of opening $2b$. Consequently, equation will, in most cases, yields a correct order of magnitude for the hydraulic opening of the largest active fractures. For pipe conduits of length $L$ (Fig. 8C), the air effective transmissivity is function of the fourth power of radius and, similarly, the larger conduit will dominate the flow. According to Poiseuille's law the hydraulic radius $r$ of a pipe conduit may be defined as:

$$r = \left(\frac{8}{\pi}\lambda_a L\right)^{\frac{1}{4}},\tag{16}$$

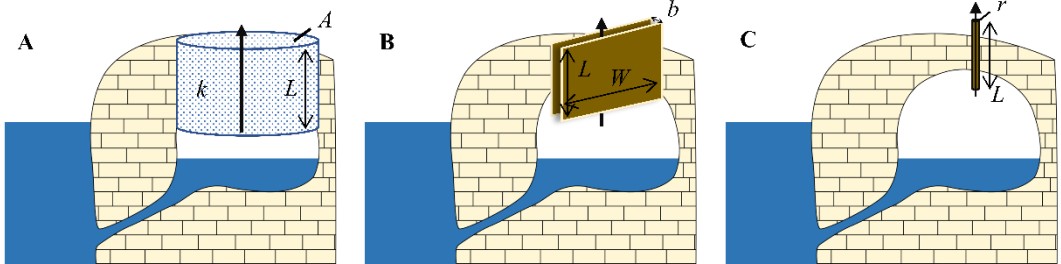


**Figure 8: Schematic cross-sections to illustrate the three theoretical models tested for flow: (A) porous rock volume of cross-sectional area A and length L (B) fracture of length L, width W and hydraulic aperture b and (C) pipe of length L and radius r.**

**5 Results**

**5.1 Cosquer cave volume**

Volume calculations are done over July and August for years 2015 to 2020, using pressure and temperature variations between successive tidal extrema. These two months were chosen during the summer season, when the cave water level is mainly driven by the tide, without significant pressurization event. This yields 4 volume calculations per day and 240 values for both months.

Figure 9A shows for years 2017 and 2018, all calculated values taking into account the measured air temperature variation

during the tide (referred to as temperature corrected volume, Eq. (5)), with measurement uncertainties. The volume of the cave and its uncertainty are computed using the weighted mean (Eq. (B5)) and weighted mean standard deviation (Eq. (B6)) over July to August period. This method gives more weight to values with smaller uncertainties. Using data recorded in summer 2017, mean cave volume is $4973 \pm 83$ m$^3$, for a mean cave water level $\overline{h_w} = 1.60$ m$_{sw}$ (water level reference is the





absolute location of the sensor as shown in Figure 3A). Using data recorded in summer 2018, mean volume is $4967 \pm 78$ m³
for a mean cave water level $\overline{h_w} = 1.53$ m$_{sw}$, which gives 4915 m³ for an equivalent water level of $h_w = 1.60$ m$_{sw}$ in order to be
compared with results of 2017. The difference of 58 m³ between 2017 and 2018 is within the range of uncertainties.

Table 2 summarizes mean cave water level measurement and volume calculated for summers between years 2015 and 2020.
Mean cave volume over the 6 years is 5000 m³ for an average water level of 1.54 cm$_{sw}$. Maximal mean cave volume is
5164 m³ in 2020 when mean water level is minimal (1.40 m$_{sw}$) and is minimal in 2016 with 4957 m³ when mean water level
is maximal (1.64 m³). Considering the entire time series, the annual cave average water level for years 2015 to 2020 is
1.33 m$_{sw}$ and the annual mean volume of the cave on this 6 years-period is 5184 m³.

|  | 2015 | 2016 | 2017 | 2018 | 2019 | 2020 |
|---|---|---|---|---|---|---|
| $\overline{h_w}$ (m$_{sw}$) | 1.55 | 1.64 | 1.60 | 1.53 | 1.55 | 1.40 |
| $\overline{V}$ (m³) | 4974 | 4957 | 4973 | 4967 | 4965 | 5164 |

**Table 2: Mean cave water level and volume over July and August for years 2015 to 2020**

In order to show the impact of the heat exchanges between the air and the cave walls or water pools on the cave volume
calculation, we performed calculation for three assumptions on year 2017 data, plotted in Fig. 9B. Case I is the temperature
corrected curve (red line on Fig. 9B), using air temperature variation during the tide (Eq. (5)); this is the curve connecting the
data point given in Fig. 9A. Case II (green dashed line on Fig. 9B) is the volume calculation with an isothermal assumption
(Eq. (6)), and case III (blue line on Fig. 9B) is the volume calculation with an adiabatic assumption (Eq. (8)). Comparison of
cases I, II and III over July and August 2017 shows that the three cases give similar relative variations but different mean.
Temperature (I) and isothermal (II) plots are almost superimposed, suggesting that the isothermal assumption is a much
better approximation than the adiabatic one. The small tide-related temperature variations in the cave (< 0.05 °C) shows a
quasi-isothermal process, which leads to a mean relative difference less than 1 % with isothermal computation. Mean
volume for adiabatic case is 6993 m³. There is a factor of 1.4 between case III and case II because of the adiabatic index γ
(Eq. (8)). This gives the possible maximum cave volume in the case when the available temperature record would not be
representative of average air temperature variations in the cave during the tidal cycle.





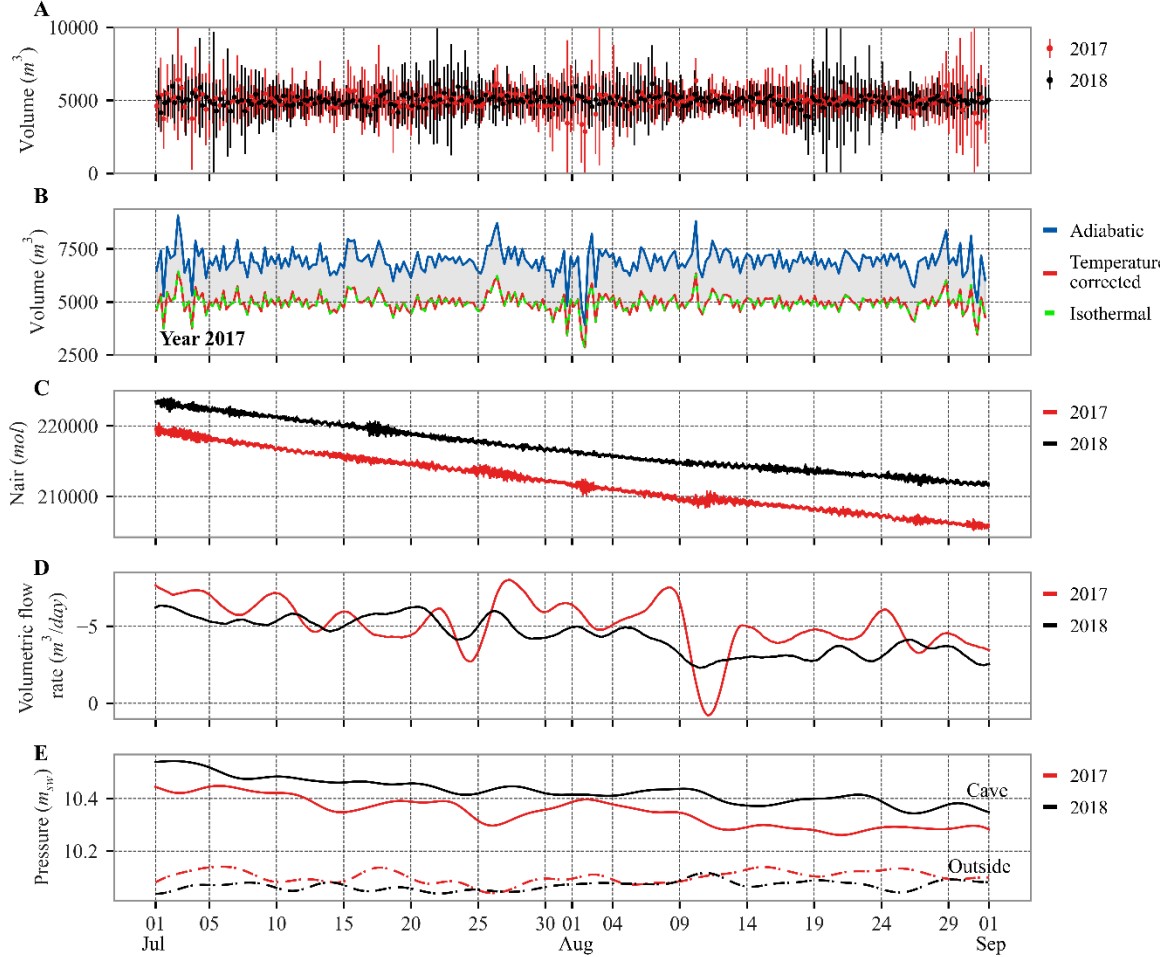


**Figure 9: Calculated and recorded time series from July, 1 to August, 31 for years 2017 and 2018. (A) Results of the cave volume computation using air temperature variations, with uncertainties (B) Results of the cave volume computation (only year 2017) for the three assumptions: adiabatic, isothermal, and temperature corrected (using air temperature variations); (C) Results of cave air quantity; (D) Results of volumetric flow rate (negative value for outflow) filtered with a 5 days Hann window; (E) Recorded data**
**of cave air pressure and atmospheric pressure (outside the cave) filtered with a 5 days Hann window.**

**5.2 Cave airflow rates**

**5.2.1 Outflow during slow pressure decrease**

The cave volume previously calculated makes it possible to evaluate at any time the cave air quantity (Eq. (9)) and therefore the airflow rates (Eq. (10)) for standard pressure/temperature conditions ($P_0$ =101325 Pa, $T_0$ = 288.15 K). The air quantity
given in Fig. 9C shows a slow decrease over the two months, correlated with the slow air pressure decrease (Fig. 4). Quantity of air is lower in 2017 than in 2018 because the air pressure was lower in summer 2017 in the cave. Table 3 summarizes mean airflow rates over July and August for years 2015 to 2020. Values are negative when air flows out of the





karst. Mean summer flow rate over the 6 years is -5.9 m³ d⁻¹ (-0.24 m³ h⁻¹), ranging from a minimum -4.5 m³ d⁻¹ (in 2018) to a maximum -7.7 m³ d⁻¹ (in 2020). Years 2017 and 2018 are detailed in Fig. 9D, filtered with a 5-days Hann window. Airflow
rates change from -1.2 to -8.4 m³ d⁻¹ in 2017. These extrema values appear mainly when air quantity curve (Fig. 9C) shows a noisy shape, corresponding to periods with higher sea level variations (waves outside the karst). In 2018, airflow rates were less varying, corresponding to a smoother decrease of air quantity over the summer.

|  | 2015 | 2016 | 2017 | 2018 | 2019 | 2020 |
|---|---|---|---|---|---|---|
| Q (m³ d⁻¹) | -5.2 | -5.7 | -5.3 | -4.5 | -6.8 | -7.7 |

**Table 3: Mean cave airflow rates over July and August for years 2015 to 2020 (air flows out of the cave for negative values)**

Nevertheless, in 2017 and 2018, the airflow rates tend to decrease (from the highest negative value to lowest negative value)
over the summer as the air overpressure inside the cave decreases (Fig. 9E) and therefore the pressure difference between the cave and the outside atmosphere decreases.

**5.2.2 Inflow and outflow during pressurization events**

The method to calculate the airflow rate, based on cave air quantity variation, can also be applied to net airflows during pressurization events, which occur during wintertime (Fig. 4). The pressurization event presented in Fig. 5 (from 29/04 to
01/05/2018) has been separated in several stages using pressure variations. To enhance precision, the onset of the pressurization event is identified by analyzing changes in air quantity, thus avoiding tidal variations passing point A to A' (Fig. 5). For this event, considered as a typical example, the rising limb, from the beginning of the increase of the air quantity to the peak, lasts 9.5 hours for a total inflow of 869 m³ (from A' to D). Then the rapid falling limb, from the peak to the first inflection point, lasts 30.5 hours for a total outflow of 656 m³ (from D to E). The maximum pressure increase during
the injection stage was related to a maximum inflow rate of 222 m³ h⁻¹ (from B to C). All the results are summarized in Table 4. During this pressurization event 26 % of the total air volume is injected in about 10 % of the total pressurization stage duration (from A' to D). The mean outflow rate during the rapid pressure drop (from D to E) is 82 times higher than the mean airflow rate during the slow pressure decrease in summer season (July and August, mean from 2015 to 2020). 75 % of the air injected during pressurization stage (from A' to D) leaked out of the cave in the next 30.5 h (from D to E).

|  | Stage | Duration (h) | Airflow rate (m³ h⁻¹) | Total volume (m³) |
|---|---|---|---|---|
| A' → D | Rising limb | 9.5 | 91.7 | 869 |
| B → C | Maximum slope | 1 | 222.6 | 222 |
| D → E | Rapid falling limb | 30.5 | -21.5 | -656 |

**Table 4: Airflow rates during the pressurization event from 29/04 to 01/05/2018**

This method is applied to quantify the total net volume of air flowing in and out of the cave during all the pressurization events spanning from 2015 to 2020. The cumulative annual net air inflow ranges from 10240 m³ (year 2015) to 22460 m³ (year 2020) with an annual average of 17590 m³. Similarly, the cumulative net annual air outflow during rapid pressure decays varies from 7720 m³ (year 2015) to 18260 m³ (year 2020) with an annual average of 13270 m³. This yields an annual





average sum of 4300 m³ for the volume of air flowing out of the cave during slow depressurization periods. These results of net volume flowing in and out of the cave will be used to discuss the air renewal in section 6.2.

## 5.3 Permeability of the limestone massif

During the periods of slow pressure decrease in July and August, the air effective transmissivity coefficient is calculated according to Eq. (13). It is then converted to effective permeability, effective hydraulic aperture and effective radius,

corresponding to different ideal flow geometries, as defined in section 4.4.

1 – It is firstly assumed that air flows out through a porous rock volume of cross-sectional area $A$ and length $L$ to compute its air effective permeability $k_a$ (Eq. (14), Fig. 8A). We consider two end member cases consistent with the Cosquer cave geometry: i) flow through the rock volume above the "Grand Puits", hence $A = 100$ m² and $L = 10$ m ($A/L = 10$ m), and ii) flow through the whole rock volume above the main cave, hence $A = 2000$ m² and $L = 40$ m ($A/L = 50$ m). All the averaged

results between 01/07 and 31/08 for the years 2015 to 2020 for the different models are summarized in Table 5, varying between 4.6 $10^{-15}$ to 50.0 $10^{-15}$ m².

2 – It is assumed that air leakage occurs through a fracture of length $L$ and width $W$ to determine its hydraulic aperture $b$ (Eq. (15), Fig. 8B). Ratio $L/W$ is set at 1. Results vary between 0.14 mm and 0.18 mm (Table 5).

3 – It is assumed that air flows out through a small karst conduit equivalent to a pipe of length $L$ and radius $r$ (Eq. (16), Fig.

8C). Two cases are set up: i) the pipe goes from the main cave to the surface, hence $L = 40$ m, and ii) the pipe goes from the top of the "Grand Puits" to the surface, hence $L = 10$ m. The computed radius of this hypothetic pipe is then around 1.5 to 2.7 mm (Table 5).

| | | 2015 | 2016 | 2017 | 2018 | 2019 | 2020 |
|---|---|---|---|---|---|---|---|
| $k_a$ ($10^{-15}$ m²) | $A/L = 10$ m | 29.9 | 50.0 | 40.3 | 23.3 | 32.3 | 29.5 |
| | $A/L = 50$ m | 6.0 | 10.0 | 8.1 | 4.6 | 6.4 | 5.9 |
| $b$ (mm) | $L/W = 1$ | 0.15 | 0.18 | 0.17 | 0.14 | 0.16 | 0.15 |
| $r$ (mm) | $L = 10$ m | 1.66 | 1.88 | 1.77 | 1.55 | 1.69 | 1.65 |
| | $L = 40$ m | 2.33 | 2.66 | 2.50 | 2.20 | 2.39 | 2.34 |

**Table 5: Averaged results over July and August for years 2015 to 2020 of the air effective permeability k_a considering 2 different geometries, the equivalent permeability of a fracture with a hydraulic aperture b and the equivalent permeability of a pipe of**

**radius r considering 2 different pipe lengths.**

## 6 Discussion

### 6.1 Exploring cave with sea tide: access the inaccessible volumes

Cave volume is a challenging parameter to get and is important as it is involved in the study of the cave air renewal or internal air flow impacting conservation of work art or archaeological remains. It can also be a parameter of interest for



archaeologic studies, to understand the spaces that paleolithic human used for decoration. Indeed, the large number of Upper
      Paleolithic caves decorated in south of France shows that volumes of decorated caves vary in a wide range, from small
      rooms to large caves. For instance, Lascaux cave volume is lower than 3000 m³ (Malaurent et al., 2006), the polychrome
      room in Altamira cave is 342 m³ (Sainz et al., 2018), the Chauvet cave is 60000 ± 20 000 m³ (Bourges et al., 2020b), Cussac
      cave is 50000 m³ (Peyraube et al., 2016) and l'Aven d'Orgnac is 237000 m³ (Bourges et al., 2006a). Volume of accessible
parts of caves had been usually obtained by 3D speleological survey (Jouves et al., 2017), using handheld topographic
      instruments or by laser scanning (Giordan et al., 2021; Mohammed Oludare and Pradhan, 2016). These methods are efficient
      but are mainly limited by the accessibility of cave passages to speleological investigation because they required to scan point
      clouds. Small passage or connected rooms yet undiscovered are then not surveyed. In the case of the Cosquer cave, tidal
      pressure variation makes it possible to assess the entire cave volume independently the geometry or the accessibility of these
volumes. The method uses classical equations but is a kind of exceptional application since partly drowned confined coastal
      caves are not widespread. Nevertheless, calculating the whole volume of the cave gives opportunity to compute other crucial
      data in this study: air quantity and then air flow rates and rock permeability. One strength of the method is that it uses the
      natural variation of the pools water level induced by the sea tides 2 times per day, giving four slopes per day. Results
      uncertainties vary with tide range. Uncertainties are maximum when the tide variation is minimum. Computing the mean
volume over a large time (2 months) is a way to minimize the impact of local disturbances (Fig. 9C), generating a few
      outliers in volume results, although the sources of the disturbances have not been identified and separated. One should
      remember that the cave volume is almost constant from day to day, and volume changes in Fig. 9A from one result to the
      next are commonly explained by bias in the measurement (sensor dependent), or due to water level variations with a very
      high frequency when the seawater is moving with waves.

On a two months scale, there is a slow decrease of the cave air pressure and the pools water level rises slowly in the cave.
      The cave volume should then change, decreasing with the rising of the water level. As shown in Fig. 4, between July and
      August, water level rises by 23 cm$_{sw}$ in 2017 and 13 cm$_{sw}$ in 2018. Using the reference pools surface (847 m²), the cave
      volume decrease is equivalent to 192 m³ in 2017 and 112 m³ in 2018 during these two months. No significative decreasing
      trend appears on the cave volume measurement time series (Fig. 9A). However, a significant difference in average volumes
(197 m³) is found between year 2020 when the water level remained exceptionally low (1.4 m) than during years 2015-2019
      (average water level 1.57±0.05 m) (Table 2). We also compared this computed volume, with the volume calculated from the
      3D survey of the cave. We approximated roughly the volume of the cave from the 3D speleological hand-survey maps and
      cross-sections (Fig. 2): 3100 m³ for the decorated rooms (#1 in Fig. 10), 1500 m³ for the "Grand Puits" (#2 in Fig. 10) and
      1200 m³ for the upper conduit connected to the top of the Grand Puits (#3 in Fig. 10). The total estimated volume is 5800 m³,
which is significantly higher than our best estimation (5184 m³ for water level of 1.33 m). Assuming the geometrical
      determination is accurate, which still needs to be confirmed by full 3D modeling of the whole cave, this difference may be
      explained by the actual surface of the pools which will need to be refined and by the temperature variations of the air during
      a tidal cycle. Regarding air temperature, convective movements control temperature homogenization in the cave (Lismonde,



2002) and although the air temperature time series were obtained outside the convective boundary layer, they were only

acquired at one location in the main room, and may not be representative of the volume-averaged temperature in the cave. We pointed out that the recorded tide-related temperature variations were much smaller than the adiabatic tide-related temperature variations and showed that calculations in fully adiabatic conditions could result in 40% larger volumes estimates. Therefore, other parts of the cave may have larger tide-related temperature variations, which could result in underestimating the total volume. Nevertheless, our results suggest there are no other connected large rooms to discover

inside the Cosquer cave.

## 6.2 Air renewal

Cave airflows typically take place through the entrance of the cave along karst conduit or through fractures or porosity in the formation. If the cave is connected to several entrances, even too small for human investigation, ventilation occurs through the karstic network between the entrances (Gabrovšek, 2023; Lismonde, 2002). Here, in the Cosquer cave, the cave is

confined, all the passages are closed by submarine karst sumps. The air pressure inside the cave is almost always higher than the pressure outside and we showed in the results section that airflows are conditioned by the following: (1) there are rapid air exchanges driven by waves during short pressurization events occurring a few dozen times a year, with a rapid inflow and outflow through the saturated karst conduits, these events generally have a positive airflow budget which results in a net inflow of air to the cave; (2) excess air slowly leaks through the limestone walls in the unsaturated zone re-equilibrating air

pressure on a seasonal time scale. The processes allowing air renewal are thus very different from other caves, which do not have long-term overpressuring.

During the summer period, we showed that there is no significant air inflow and the air flows out continuously (Fig. 10A). So, the air quantity inside the cave varies, decreasing over the summer, but the air is not renewed. The annual mean pressure difference between inside and outside the Cosquer cave is 56 cm$_{sw}$ (years 2015 to 2020). This is significantly higher than the

natural pressure gradients usually found in caves: pressure gradients due to variations in atmospheric pressure or temperature variations in the range of 2 cm$_{sw}$ are observed at the Lascaux and Altamira caves (Houillon et al., 2017; Sainz et al., 2018) and up to 5 cm$_{sw}$ in the case of a coastal karst under the influence of tides (Jiao and Li, 2004). In the case of the Cosquer cave, this air supply and consequently air renewal, only happens during pressurization events. These events require an additional mechanism to force the air to flow from outside to inside the cave. Field observations and data suggest a close link

between sea conditions (waves height and direction) and the pressurization events. Waves breaking along the cliff at the karst inception horizon seems to be the main driver. Understanding this mechanism is beyond the scope of this paper but our air content calculations can be used to evaluated net fluxes occurring during pressurization events.

Pressurization events generally have an initial pressure increase stage indicating inflow, followed by a rapid decay indicating a net outflow through the karst, probably occurring through the same shallow conduit network. Part of the injected air is thus

expelled right after the pressurization stage. For instance, during the 29/04/2018 event (Fig. 5), 869 m$^3$, corresponding to





17 % of the cave's air volume, is injected in a few hours but nearly 75 % of this air exits during the following day (Table 4). As no air quality measurements were carried out, it is unclear whether the air exiting is newly entered air or mixed with pre-event air. Moreover, the input air may at first stay near the entrance area and not mix with air in the other rooms. This situation has been reported in many caves. Its occurrence depends on the air density differences inside the cave (Lismonde,
2002; Peyraube et al., 2016) and on the shape of the cave passages (Gabrovšek, 2023; Luetscher and Jeannin, 2004). In the Cosquer cave, in situ observations suggested that air inflow occurs mainly through the small pool connected to the upper conduit (Malaurent and Vouvé, 2003) and not directly in the decorated rooms We propose a conceptual model in Fig. 10, taking into account the volumes of the three main parts of the cave. The air pushed by the waves below the sea level flows up to pools connected to the lower end of the upper conduit (Fig. 10B). The ascending geometry of the upper conduit may
prevent air from flowing down to the decorated rooms. The volume of air flowing in during the event must therefore be greater than the volume of the upper conduit (around 1 200 m$^3$) to reach the top of the Grand Puits and flows into the decorated rooms.

Summing all the air volume entering the cave during events, the mean annual air inflow volume is approximately 17590 m$^3$ (years 2015 to 2020), corresponding to a total annual air renewal of 3.4 times the volume of the cave (considering the mean
cave volume of 5184 m$^3$ and under standard pressure/temperature conditions) per year. However, this value represents a maximum renewal rate as 75 % (13270 m$^3$) of the entering air exits during the rapid pressure decays immediately after the pressure peaks (e.g. Fig. 5, from D to E), possibly without mixing, and 25 % (4300 m$^3$) exits during slow pressure decreases. The minimal air renewal rate in the cave is calculated excluding the fraction of air outflowing during pressurization events, in other words, only taking into account the volume of air leaving by slow depressurization. This minimum rate is 0.8 times
the air volume in the cave per year (mean from year 2015 to 2020). In either case, air renewal in the Cosquer cave is significantly lower than measured at Hollow Ridge Cave (~175-8760 y$^{-1}$, Kowalczk and Froelich, 2010), in the Altamira polychrome hall (~270 y$^{-1}$, Sainz et al., 2018) or at the Chauvet cave (~40 y$^{-1}$ Bourges et al., 2020b). For preservation reasons, the Altamira and Chauvet decorated caves are artificially closed by gates to limit natural air ventilation, which nevertheless remains 11 to 300 times higher than in the Cosquer cave.

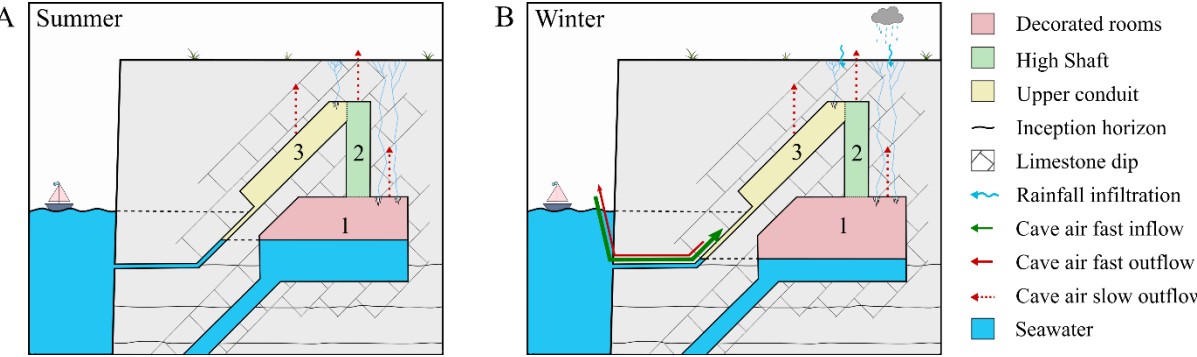


Figure 10: Conceptual model of airflows in the saturated and unsaturated zones of the limestone massif surrounding the Cosquer cave (cross-section, not to scale).



## 6.3 Permeability of the unsaturated zone at the cave scale

The case of the Cosquer cave shows that permeability can be highly variable between the saturated zone below the sea level
and the unsaturated zone surrounding the cave above the sea level. Both saturated and unsaturated zones are in the same rock
age and facies, i.e. early Cretaceous limestones with urgonian facies. At the rock massif scale, permeability differs mainly if
karst conduits are connected, or if karst fractures or porosity are filled and clogged by low-permeability materials. Study of
water level variations in coastal wells is a classic way of calculating the aquifer's transmissivity and storage coefficient in the
saturated zone, using the amplitude, frequency and phase shift of the tide (Trefry and Johnston, 1998; Zhang, 2021). This
method does not apply to coastal karsts that do not filter the tide pressure wave, when karst conduits are large. As
groundwater table fluctuates with sea tides in coastal aquifers, it also causes air pressure fluctuations in some coastal
unsaturated zones. Tide induced airflow has been extensively studied (Kuang et al., 2013). Coastal aquifers with layered
unsaturated zone have airflow induced by sea tides. Jiao and Li (2004) or Xia and collaborators (2011) used this feature to
validate air permeability estimation with numerical simulations. In the case of the Cosquer cave, air effective permeability
calculations have not been done using the tide variation because it requires pressure measurements at several altitudes in the
unsaturated zone. However, we showed that air effective permeability calculations can be done on a seasonal time scale
using the cave slow depressurization. It gives air effective permeabilities varying from $4.6\ 10^{-15}$ to $50.0\ 10^{-15}$ m$^2$ (12 values).
These values are relatively high compared to permeability given in the literature for samples taken in similar carbonate
formations. Several authors reported permeability measured on plugs (local scale, few centimeters) in Urgonian limestones
in south of France, ranging from $<1\ 10^{-17}$ m$^2$ to $4\ 10^{-15}$ m$^2$ for porosities ranging from 0.75 to 18.3 % (Cochard et al., 2020;
Danquigny et al., 2023; Jeanne et al., 2013). These measurements were done on plugs and do not include karstic vugs,
neither fractures or inception horizons on bedding planes (Filipponi et al., 2009) that increase locally the rock permeability.
For instance, larger scale measurements performed with packers in a fault zone with open fractures in the same formation
found much higher permeabilities, of the order of $6.9\ 10^{-12}$ m$^2$ (Guglielmi et al., 2015). Air effective permeabilities
determined by air depressurization of the Cosquer cave give permeability of the unsaturated zone at the massif scale, i.e. at a
mesoscale around 100 m length scale and include fractures and karst conduits. These potentially permeable geological
features have been recognized in-situ on the cliff and the plateau around the cave. However, we calculated that the
permeability of the massif surrounding the cave is equivalent to a fracture of small hydraulic aperture (1 equivalent fracture
around 150 µm), or an equivalent pipe of very small radius (between 1.5 and 2.7 mm). These relatively small values show
that most voids or fractures may be clogged or not connected in the unsaturated zone. This is consistent with in situ
geomorphological observations at the outcrop, where karst voids are generally filled up by calcite and clay minerals.





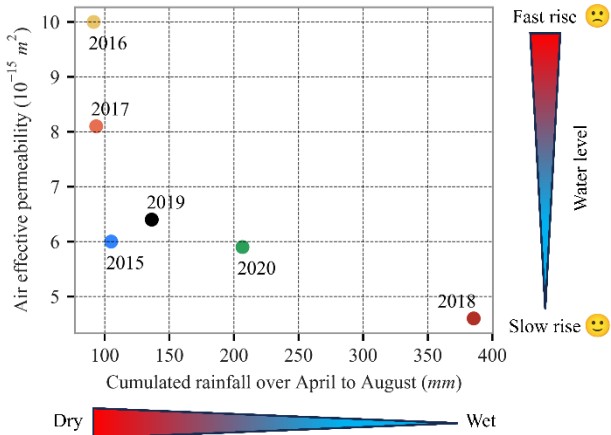

**Figure 11: Mean air effective permeability (m²) between July and August as a function of cumulated rainfall at Cassis from April to August (mm) for years 2015 to 2020 (for model A/L = 50 m)**

Fig. 11 shows the mean air effective permeability calculated for each year over July and August compared to the cumulated rainfall over April to August (5 months). We computed cumulated rainfall for several periods lasting over the slow cave air pressure decrease period, beginning on 1st of May, April or March and ending on 31 August (end of the period used for permeability calculation). The trends were similar and the best graphical view is given in Fig. 11. This figure shows a relationship between the air effective permeability of the unsaturated zone and the cumulated rainfall. There are no data

available on the moisture content in the unsaturated zone but it may be assumed that the higher the cumulated rainfall is the higher the moisture in the unsaturated zone should be. A lower water saturation in the limestone massif is then expected during dryer years, i.e. years with longer spring and summer droughts. Consequently, dryer years should have a higher relative permeability to air because there is less groundwater in contact with grains and more air connection between the pores (Kuang et al., 2013; Weeks, 1978). It may also increase the number of flow paths or open a preferential airflow path

through the karstic unsaturated zone. On the other hand, the rainfall infiltrates, fills the porosity and limits the air effective permeability of the unsaturated zone.

Cumulated rainfalls from April to August vary from 90 to 390 mm over the six years studied. Mean interannual cumulated rainfalls (1991 to 2020, Météo-France 2023) have been observed equal to 170 mm. Four years over six (2015, 2016, 2017, 2019) have a cumulated amount lower than the mean, including 3 years with drought, and 1 year (2018) was highly rainy.

The expected trend clearly happens in Fig. 11: air effective permeability of the unsaturated zone increases systematically when rainfall amount decreases. This trend is not linear along the whole range of cumulated rainfalls variations. Years 2019, 2020 and 2018 show a large increase in rainfall amount but a small decrease in air effective permeability, whereas dryer years show a larger increase in permeability. As a result, the cave air pressure decreases faster in dryer years and so the water level inside the Cosquer cave rises faster. Dryer years could lead to a total loss of overpressure in the cave.



## 7 Summary and conclusion

Airflows in decorated caves impact wall art conservation. In karst systems, air can flow through either the karst network or open and connected fractures or through the porous carbonates rocks. This paper gives the first conceptual model and quantification of airflows through the rock massif surrounding the Cosquer cave, including flows through the saturated zone and the unsaturated zone. Data show that the Cosquer cave air pressure is higher than atmospheric pressure. In response, the water level in the cave is lower than the sea level. Three types of cave air pressure variations at 3 different time scales are identified:

– (1) seasonal variations consisting in a succession of pressurization events from early autumn to late spring. Pressurization events consist in massive air inflows resulting in a remarkable increase of the cave air pressure that is immediately followed by a rapid pressure drop but with a positive budget showing an increase in air quantity in the cave;

– (2) a slow air pressure decrease from late spring to early autumn;

– (3) a daily cave air pressure and water level variations forced by sea tide.

We showed that tidal variations in cave air pressure can be used to calculate the geometric volume of air of the cave (for a given water surface area) by applying the perfect gases law at low and high tides. The mean cave volume calculated in summer is around 5000 m$^3$, which is consistent with a rough estimation made from the 3D speleological hand-survey maps and cross-sections. It seems that archaeologists have explored all the main rooms of this decorated cave.

Using the cave volume results, we calculated the air flowing in and out of the cave. The Cosquer cave is closed off by sumps in the saturated part of the karst below the sea level and by the low permeability of the rock in the unsaturated zone surrounding the cave. The high airflow rates during pressurization events (e.g. up to 222 m$^3$/h for the case studied in April-May 2018) revealed that air flows through large voids in the saturated zone of the karst. These results are a first step to study the mechanism that generates the overpressure inside the cave. Further studies will focus on the relationship between the pressurization events and the assumed link with the sea waves breaking on the cliff outside the cave. Following the first observations by Malaurent and Vouvé (2003), we confirm that air may use karst or inception horizon on bedding plane to flow a few meters below the sea level across the limestone massif. Consequently, air renewal occurs during pressurization events but at a low rate (3.4 to 0.8 y$^{-1}$). However, this air renewal may differ from one area to another: the volume of outside air injected into the cave during pressurization events is of the same magnitude order as the volume of the upper conduit (about 1200 m$^3$) and may therefore not reach the decorated rooms located in the lower part of the cave. The 3D scan of the cave will give insight of the volumes of the different areas of the cave. Moreover, part of the air flowing in is flowing out of the cave immediately after the pressurization peak. Further air quality measurements, such as radon concentration, will help to show whether the air exiting is newly entered air or mixed with pre-event air (Fernández et al., 1986; Cigna, 2005; Kowalczk and Froelich, 2010). In addition, excluding the periods with pressurization events, the gradual decrease of the air pressure over the summer is explained by the slow air outflow through the unsaturated zone, which does not necessarily involve air renewal.



Using the low rates of air outflowing during the slow pressure decreases (around 6 m³/d), we estimated the air effective permeability of the rock massif with the Darcy's law for several assumptions of flows through the unsaturated zone. Rock
effective permeability was found in the range $4.6 \times 10^{-15}$ to $50.0 \times 10^{-15}$ m² depending on the assumed geometry. Although karstified fractures are visible on the surface of the limestone massif, these are sealed by calcite or clay minerals, which limits permeability. This is in line with the observation of a permanently over-pressurized cave. We showed that the massif mean air effective permeability over the months of July and August varies from year to year as a function of the water saturation of the unsaturated zone by comparing effective permeability with the cumulative rainfalls over spring and
summer. A decrease in rainfall leads to an increase in the air permeability of the massif, and therefore to an increase in the air outflow rate through the unsaturated zone of the massif.

By this first six years of measurements in this decorated cave, we show that droughts periods and dryer spring and summer years impact the conservation of the paleolithic painting and engraving on the walls close to the pools water level. This result is of utmost importance in the frame of the current climate change. In the future, increasing evapotranspiration rate and
longer dry periods are expected in the Mediterranean basin (Cramer et al., 2020). Within these conditions, pools water level of the Cosquer cave should then rise faster during spring and summer, and hence decorated walls and archaeological artefacts on the floor would face longer flooding period. The effect may be even worse if clay infillings in karst voids start drying and cracks open across the unsaturated zone, transforming the current slow air pressure decrease in rapid decrease. Nevertheless, predicting water level in the future in the Cosquer cave remains hard because other parameters drive the
variations, such as the process of air inflow through the saturated zone. The striking example of the Cosquer cave also shows that inland (not coastal) decorated cave could be affected by changes in air circulation through the unsaturated zone in the future in a climate change context, which will affect the water content in the unsaturated zone governing the air permeability. Its supports the need to observe our changing world by dedicated data acquisition (Gaillardet et al., 2018).

## Appendix A : tide filter

As seen above, cave air pressure and water level vary with tides. We use this behavior to calculate the variation of the volume of air in the cave. This calculation is done at high and low tide to maximize the amplitudes of these variations and minimize measurement uncertainties. To determine the times of the tidal peaks, the tidal signal $h_t$ is isolated from the sea level $h_s$ using the TTide python package (Pawlowicz et al., 2002). The synthetic tide signal $h_t(t)$ is then derived:

Tide is low when :

$$\frac{dh_t(t_{lt})}{dt} = 0 \quad \text{and} \quad \frac{dh_t(t_{lt}-1)}{dt} < 0 \,, \tag{A1}$$

And tide is high when :

$$\frac{dh_t(t_{ht})}{dt} = 0 \quad \text{and} \quad \frac{dh_t(t_{ht}-1)}{dt} > 0 \,, \tag{A2}$$

Where $t_{lt}$ and $t_{ht}$ are time of low tide and high tide.



**Appendix B : uncertainties calculation**

Propagation of uncertainty provides the standard deviation for a sum or difference of measured parameters:

$$\sigma_f = \sqrt{\sum_{i=1}^{n} \sigma(x_i)}\,, \tag{B1}$$

And for produce and division:

$$\sigma_f = |f| \sqrt{\sum_{i=1}^{n} \left(\frac{\sigma(x_i)}{x_i}\right)^2}\,, \tag{B2}$$

Where $f$ is the resulting value, $x_i$ is the measured parameter and $\sigma(x_i)$ its standard deviation. Standard deviation of
parameters $x_i$ is given by:

$$\sigma(x_i) = \frac{\delta_i}{\sqrt{3}}\,, \tag{B3}$$

With $\delta_i$ the typical uncertainty of the sensor $i$. Using Eq. (C1) and Eq. (C2) the typical uncertainty for the cave volume $V_l$
(Eq. (5)) is:

$$\sigma_{V_l} = V_l \sqrt{\left(\frac{\sigma(\Delta h_w)}{\Delta h_w}\right)^2 + \left(\frac{\sigma(P_{ah})}{P_{ah}}\right)^2 + \left(\frac{\sigma(T_{al})}{T_{al}}\right)^2 + \left(\frac{u(K)}{K}\right)^2}\,, \tag{B4}$$

Where:

$$\left(\frac{u(K)}{K}\right)^2 = \left(\frac{P_{ah}T_{al}\sqrt{\left(\frac{\sigma(P_{ah})}{P_{ah}}\right)^2 + \left(\frac{\sigma(T_{al})}{T_{al}}\right)^2} + P_{al}T_{ah}\sqrt{\left(\frac{\sigma(P_{al})}{P_{al}}\right)^2 + \left(\frac{\sigma(T_{ah})}{T_{ah}}\right)^2}}{P_{ah}T_{al} - P_{al}T_{ah}}\right)^2$$

and

$$\sigma(\Delta h_w) = \frac{\sqrt{2\left(\sigma(P_a)^2 + \sigma(P_w)^2\right)}}{\rho_{sea}\, g}$$

The use of the weighted mean reduces the contribution of results with high uncertainties:

$$\bar{V} = \frac{\sum_{j=1}^{n} \frac{V_{lj}}{\sigma^2 + \sigma_{V_l j}^2}}{\sum_{j=1}^{n} \frac{1}{\sigma^2 + \sigma_{V_l j}^2}}\,, \tag{B5}$$

Where $\sigma$ is the standard deviation of $V_l$. The weighted variance of $\bar{V}$ is given by:

$$\sigma_{\bar{V}}^2 = \frac{\sum_{j=1}^{n} \frac{1}{\sigma^2 + \sigma_{V_l j}^2}}{\left(\sum_{j=1}^{n} \frac{1}{\sigma^2 + \sigma_{V_l j}^2}\right)^2}\,, \tag{B6}$$

Volume result is within the range:

$$[\bar{V} - \sigma_{\bar{V}}\,;\, \bar{V} + \sigma_{\bar{V}}]\,, \tag{B7}$$



*Code and data availability* : The codes developed for this research and code to reproduce the figures are available via gitlab at https://gitlab.osupytheas.fr/hpellet/chapitre-3-mesoscale-permeability-variations. Port-Miou observatory dataset can be accessed via https://data.oreme.org/snokarst/snokarst_map. Meteorological data can be requested via https://donneespubliques.meteofrance.fr/ with fees. Data from the Cosquer cave are confidential.

*Authors contribution* : HP, BA and PH conceptualized the research goals and developed the methodology. BA and ST found the funding for the project. HP developed the code and prepared vizualisation and GG provide programming support and analysis tools. HP prepared the original draft with contribution from BA, PH and ST.

*Competing interest* : The authors declare that they have no conflict of interest.

### Acknowledgements

We would like to thank the DRAC PACA (Service Régional de l'Archéologie, Conservation Régionale des Monuments Historiques), which supported this study by authorizing access to the cave and funded the fieldwork and some of the measurement devices. This study was also supported by the Aix-Marseille University (AMU) and the Laboratoire de Recherche des Monuments Historiques (LRMH, Ministère de la Culture). This work received support from the French government under the France 2030 investment plan, as part of the Initiative d'Excellence d'Aix-Marseille Université - A*MIDEX - Institute for Mediterranean Archaeology ARKAIA (AMX-19-IET-003). Hugo Pellet has been awarded a PhD grant from AMU and LRMH. This work was performed within the framework of the Port-Miou observation site, part of the KARST observatory network ([www.sokarst.org](www.sokarst.org)) initiative from the INSU/CNRS, which aims to strengthen knowledge-sharing and promote cross-disciplinary research on karst systems. This study was also performed in the frame of the interdisciplinary research team Equipe Grotte Cosquer supervised by C. Montoya (PCR 2022-2023). We thank the Conservatoire du Littoral, the Calanques National Park, the Cassis city and Météo-France. This paper benefited from the work of Bertrand Chazaly on the 3D scan of the cave, Caroline Font on QGIS and Sylvain Rassat on the 3D model. We thank the Immadras diving team, directed by Luc Vanrell and Orsane Vanrell, for their help in field studies. We also thanks Sophie Viseur, François Fournier, Philippe Léonide, Baptiste Suchéras-Marx, Lamarche Juliette, Paul Namongo Soro (AMU), and Luc Vanrell and Michel Olive (DRAC, Immadras), for the enriching conversations.

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
