# Peer review of "Mesoscale permeability variations estimated from natural airflows in the decorated Cosquer Cave (SE France)"

_EGUsphere, 2023_

## Referee Comment (RC2)

[referee-annotated manuscript omitted]

---

## Author Comment (AC1)

**Reply to reviewer 1 in blue**

Understanding the processes driving the ventilation dynamics in decorated caves is key to ensure their conservation as even minor shifts in the cave's climatic equilibrium may lead to microbial outbreaks deteriorating the artwork. Pellet et al. present a six-year monitoring study from Cosquer cave, a major paleolithic site on the Mediterranean coast. The only access to the cave is a flooded passage, thus prohibiting any major exchange with the outside environment. The study encompasses air pressure, water levels and cave air temperature. The authors confirm that the cave air pressure is always higher than in the outside atmosphere keeping the water level below the main paintings. Analyzing the effect of tides on the cave air pressure, the authors calculate a cave volume of c. 5000m3. Although largely isolated from the outside, short pressurization events during the winter season reveal sporadic ingress of external air. These events are used to assess the permeability of the host rock and, thus, may serve as reference for low permeable karst volumes in a broader context.

The paper is generally well-written although a cross-check by a native speaker would be recommended (cf. punctuation and use of articles); the figures are of good quality.

First, we would like to express our sincere appreciation to the referee for his thoughtful review and constructive feedback. In the remainder of this document, we aim to address the referee's concerns by responding point by point to his comments and questions.

However, a few points need clarification before being acceptable for publication:

**Water level**: much of the discussion relates to the air pressure in the main chamber. A key point here is that the higher pressure keeps the water level below the mean sea-level. However, no information is given about how the elevation of the water table inside the cave was determined. Is there an altimetric survey available? If so, please indicate uncertainties. Also, what is the source of the temperature fluctuations measured in the cave air? Does it correlate to the seawater temperature and are these temperature fluctuations compensated in the pressure analysis ?

The pressure sensors in the cave are intercalibrated with atmospheric pressure outside the cave ±1 hPa each time data is retrieved from the sensor (more or less every 6 months) using an additional pressure sensor that is brought in the cave in a water-tight and pressure resistant container. When the same pressure is recorded outside and inside the cave the water level is thus the same ± 1 cm.

To investigate the source of air temperature variations within the cave, we will incorporate new data in Figure 4 of the manuscript (Figure 1 in this reply): cave water temperature (see . Temperature is measured with the same probes that measure the cave water level variation. Fig. 1C shows that the air temperature variations are mainly driven by the water temperature variations. Water temperature variations are presumably related to exchange with seawater outside the cave through conduits. Air temperature variations are smoother than water temperature variations, and delayed because of heat exchange with the cave walls, which have thermal inertia.

**Short-pressurization events**: the authors associate short pressurization events with wave activity, however without providing any quantitative data with respect to this wave activity. An alternative interpretation would relate the pressurization events to aquifer recharge after rainfall episodes. Although the author state that addressing more closely the effect of sea waves is out of scope of this paper it would make sense at least to address the effect of hydrology. In absence of surface runoff one would expect that the water infiltrates into the subsurface (as shown on Fig.10), thus pushing interstitial air towards the cave. Accordingly, it would be helpful to plot rainfall distribution together with the effective hydrological recharge along the year and discuss correlations (?) with high-frequency pressure changes in the cave.

As the manuscript is focused on the permeability of the limestone massif, we initially preferred to not delve into the origin of pressurization events. However, as pointed out by both referees, the pressurization of the cave is an intriguing phenomenon that requires some explanation. To clarify this point, additional information is provided to show that pressurization events occur when there are waves on the cliff and are not correlated to rain infiltration.

First, it must be clarified that expeditions in the cave are never scheduled during rough weather, and it is forbidden to camp overnight inside the cave. That makes direct observation of pressurizing events nearly impossible. However, records show a strong correlation between episodes of high waves and pressurization events.

To support our assumption of a causal relationship between high waves and pressurization events, the significant height of the waves in front of the cave (data provided by the French Naval Hydrographic and Oceanographic Service, SHOM) and the daily rainfall will be added to Figure 4 in the manuscript (Figure 1 in this reply). While storms are often associated with rainfall, events with high rainfall but no high waves occasionally occur (e.g. August 2018) and they do not cause pressure variations in the cave. On the other hand, some pressurization events are not associated with heavy rainfall, but they are systematically associated with waves. Ongoing statistical analysis suggests a wave height threshold of about 1 m, but this threshold also depends on other factors (sea level, wave direction…). This will hopefully be the subject of a separate publication.

**The conceptual model** summarized in figure 10 does not match with some of the statements and interpretations made along the paper. In particular, the justification for an upper pathway several meters below sea-level is unclear and inconsistent with Figure 2B where the latter is placed at the sea-level. How would waves at the sea-surface propagate into this underwater gallery? Also, Fig 10 suggests there are some water inlets draining from the surface. This makes sense but again raises the question of hydrological recharge on the cave air pressure.

As pointed out by both referees, we will add at the beginning of the paper, in section 2 (after line 99) more explanation about the several paths (or levels) of connection between the sea and the cave through the limestone massif. We will redraw Figure 2B to show that the upper pathway is below the sea-level at a deepest level (so not "placed at the sea-level" as reported by referee 1), and redraw in Figure 10 (conceptual model) the upper pathway at a shallow level (still below the sea-level of course). Given the existence of karst pathways at several levels at the beginning of the paper will help the reader to conceptualize that waves can generate bubbles and then force seawater and air to flow through the limestone massif. The detailed mechanism of how bubbles of air can flow inside the submerged karst is out of the scope of this paper. At the present state of the research, we focus in this paper on the pressure data, and by adding data of wave height and rainfall (previous comment) it should clarify that air enters the cave by the sea.

l.41: underline that the stronger airflow in winter is valid for a descending conduit. In ascending conduits, this would likely happen in summer.

Modified sentence: "These flows are subject to seasonality, with generally stronger flows in winter and stratification of air masses in summer *for descending conduits and conversely for ascending conduits*"

l.63: please state the supposed mechanism driving this airflow. Obviously, this wouldn't happen without an external force if the cave is over-pressured.

This comment is linked with the previous general comments. In the introduction section, line 63, we will introduce the supposed mechanism. "Cave air pressure increases by the inflow of outside air, during periods with high waves on the cliff. Waves can produce and force air bubbles to propagate by submarine open fissures or karst conduits."

l.91: Cretaceous should be capitalized

Corrected

l.92: "thin sections observations" delete observations

Corrected

l.134: "monitoring" instead of "survey"

Corrected

l.189: tide-related temperature variations "in the cave air" (?)

Added *in the cave*

l.195: phrasing: According the cave air temperature measurements it is rather the wall which stays close to equilibrium with the cave air rather than the opposite.

Rephrased : *"These observations indicate that thermal convection is very active at least in the decorated rooms of the cave and that at the tide time scale, the air in the cave and the walls remain close to thermal equilibrium."*

There is no direction implied here.

l.201: Fig 6 suggests that, during low tide, the water level is higher in the cave than at Port Miou. Can you expand on this?

It is because water levels are centered by their mean, see l.176: […] *pressures expressed in meter of seawater ($m_{sw}$) and mean-centered at a two days scale* […]

l.205: In Fig 6, you show that the cave water level is c. 5 cm lower than the sea level whereas the fluctuations shown on Fig 7 range in the order of 1 m. What do I get wrong?

Fig. 6 compares tide-related water level and sea level variation (around few centimeters) whereas Fig. 7 presents the annual amplitude of cave water level (up to 1.5 $m_{sw}$). We will clarify this.

Fig.7: I think I get the point but you may want to explain better why the sea-level was c. 0.25 cm higher in 2018 than 2017. What do the horizontal lines (red and black) represent on the figure?

Fig. 7 only shows the cave water level (not sea-level). To clarify, we will add line 203: "The highest level recorded in 2017 (water level close to 1,8 meters above the probe in Figure 7) was an exceptionally high water level in the cave, equivalent to the sea-level for a few days. A scaled photo of the Horses panel is added to the figure. It illustrates how high the water level can rise and flood the artwork."

The horizontal lines represented the lowest water level reached respectively in 2017 and 2018. It is confusing, the horizontal lines are then removed.

l.241: This variation is

Corrected

l.290: where does the factor 2 on the right hand-side come from?

It is a fundamental property of the gradient operator: $grad(xy) = x\, grad(y) + y\, grad(x)$ or in the case $x = y$, $grad(x^2) = 2x\, grad(x)$ (Lang, 1999).

l.307: what drives the temperature variations in the cave air? Using a constant Trock is an approximation as we know there are seasonal fluctuations in both, the cave air and the outside atmosphere? what is their effect?

Dealing first with the outside temperature variation. Considering an annual temperature oscillation of $\Delta T = 20\,°C$ for outside air (at the local meteorological station Cassis, https://www.data.gouv.fr/fr/datasets/r/5669e5be-8c67-4fbf-aeae-08cbc2369dd4). Considering the rock massif above the cave is a semi-infinite medium: in this case, the temperature oscillation diminishes with depth like: $e^{-z\left(\frac{\sqrt{\omega}}{2\kappa}\right)}$ (Carslaw and Jaeger, 1959). Where $\omega = \frac{2\pi}{t}$ is the pulsation, $t$ is the period ($t = 1$ year), $z$ is the depth and $\kappa = 0.606\ 10^{-6}$ m$^2$/s is the limestone thermal diffusivity. At 10 m depth (approximatively the depth of the top of the Grand Puits), the heat wave is diminished by a factor 0.017 and rock temperature variation due to the outside temperature oscillation is 0.34 °C.

Second, temperature variation affects dynamic viscosity of air therefore the massif permeability to air. For an ideal gas, dynamic viscosity $\mu(T)\sim\sqrt{T}$ and $\mu(T) \approx \mu'\sqrt{\frac{T}{T'}}$ (Chapman and Cowling, 1990), therefore permeability $k(T)\sim T^{\frac{3}{2}}$ and $k(T) \approx k'(T)\left(\frac{T}{T'}\right)^{\frac{3}{2}}$, where $\mu$ (Pa.s) is the dynamic viscosity and $k$ (m$^2$) the permeability at the temperature $T$ (K) and $\mu'$ the dynamic viscosity and $k'$ the permeability at $T'$. Assuming in the worst case that rock temperature ranges the same as cave air temperature (from 16°C to 21°C), then permeabilities would varies of a factor about 0.990 (at 16°C) to 1.015 (21°C). We may explain that in the revised manuscript, but doubt it is necessary as the corresponding uncertainty is much smaller that the uncertainty on the air flux measurement.

l.325: please edit: 'equation will, in most cases, yields …'

*Corrected: "In most cases, eq. (15) yields a correct order […]"*

l.345: unclear, please edit

*Edited: "Using data recorded in summer 2018, mean volume is 4967 ± 78 m³ for a mean cave water level observed $\overline{h_w}$ = 1.53 m$_{sw}$., which would be equivalent to 4915 m³ for the reference level h$_w$ = 1.60m. Using this latter value, calculated air-filled volume of the cave is the same in 2017 and 2018, within the range of uncertainties, as expected. It shows that using data of the two summer months, volume can be calculated by the method proposed."*

l.350-351: unclear, please edit. How does this compare with figures in the previous sentence?

*Edited : "Applying Eq. 3, the average annual volume of the cave can be estimated from the average volume of the cave and the average water level during summer (V = 5000 m³ and h$_w$ = 1.54 cm$_{sw}$). The average water level for years 2015 to 2020 is 1.33 m$_{sw}$ and yields an average volume of the cave of 5184 m³."*

l.380: missing article: when the air

*Corrected*

l.405: unclear, please rephrase: are your referring to the rapid pressure decay or the slow depressurization.

*Rephrased: "[…] Similarly, the cumulative net annual air outflow during rapid pressure decays varies from 7720 m³ (year 2015) to 18260 m³ (year 2020) with an annual average of 13270 m³. Therefore, air outflow outside the pressurization events can also be calculated by subtracting the cumulative annual net air inflow with air outflow during rapid pressure decays. This leads to the cumulative net annual air outflow during slow depressurization periods (4300 m³ on average)."*

l.435: have been

Corrected

l.450: you may want to specifiy that this is during summer

Corrected: "*Over the two summer months, there is a slow decrease* […]"

l.452: the pools' reference surface (?)

Corrected

l.455: should be "and during years"

Corrected

l.476: waves where not addressed yet, did they? Wouldn't it rather be associated with groundwater recharge?

See previous comments about the correlation between rainfall, waves and pressurization events

l.490: Wave heights and direction were not discussed so far

Waves heights will be addressed earlier in the manuscript (in sections 2 and 3) according to the new information provided in previous comments.

l.502: This upper conduit was not introduced yet

The upper conduit was poorly introduced l.99. The new information provided in previous comments should clarify the issue.

l.550    not connected to the cave. But what about the epikarst, flushing air into the fracture during recharge events?

See previous comments about the correlation between rainfall, waves and pressurization events

Table1  Please provide also numerical estimates and reference to it if these are used in further calculations

Known model input values e.g. $P_0$, $T_0$, $S_w$, $g$, $R$, $\gamma$, $\rho_{sea}$, $h_0$ will be added to Table 1.

Fig.2a: the 3D projection is difficult to read and a classical speleological survey would probably be more helpful here. In particular, it is unclear how P2 is connected to the rest of the chamber. Is this a small "island" in the middle of a pool? What is the grey-scale of the 3rd dimension? Adding some elevation quotes would be useful, or even better, draw some isolines.

Modification to increase readability (such as isoline instead of grey scale elevation) will be provided to Fig. 2A. The probe $P_2$ is moored in the water to a speleothem, the *island* is in fact a column.

Fig. 2b please add a vertical scale to this (nice) illustration

Vertical scale will be added

Fig.4C  Please plot also the seawater temperature

See Figure 4C updated (= figure 1C in this document)

Fig. 11 interesting figure, but one would also like to see how hydrological recharge correlates with cave Pair rises during the winter months, resp. get an idea of rainfall distribution along the year.

See Figure 4D updated (= figure 1D in this document)

[Figure]

*Figure 1: Pressure, water level and temperature time series recorded in the Cosquer cave and at the Port-Miou observatory for years 2017 and 2018: (A) Sea level at Port-Miou ($h_s$) and Cosquer cave water level ($h_w$), expressed in column of seawater ($m_{sw}$) above the probe with the same reference level. The green dashed line shows the bottom of the horses panel (paleolithic decorated wall). (B) Atmospheric pressure ($P_{atm}$) outside the cave and cave air pressure ($P_a$). (C) cave air temperature ($T_a$) and cave water temperature in Room 1 ($T_w$). (D) daily rainfall and (E) waves significant height in front of the Cosquer cave. Pressurization events periods are highlighted in grey.*

**Bibliography**

Carslaw, H.S., Jaeger, J.C., 1959. Conduction of Heat in Solids. Clarendon Press.

Chapman, S., Cowling, T.G., 1990. The Mathematical Theory of Non-Uniform Gases, 3rd edition. ed. Cambridge University Press.

Lang, S., 1999. Fundamentals of differential geometry, Graduate texts in mathematics. Springer, New York Berlin Heidelberg.

---

## Author Comment (AC2)

**Reply to reviewer 2 in blue**

The manuscript as such is interesting in many ways. It deals with the *culturally extremely important* cave and presents an interesting, unusual and *valuable set of data*. As such I think authors should be motivated to revise the paper to a publishable version.

My criticism mainly goes to the clarity of the presentation, which makes the manuscript hard to read. Some important concepts and objectives should be clearly stated earlier in the paper to keep the reader interested. For example, the pressurization events are shown early on, but their essential role -- they are the reason for the long-term overpressurization of the cave (I guess do?) -- is not told.

We would like to thank the referee for the time spent reviewing the paper and for the valuable comments. We have carefully considered the feedback and believe the new information provided in this document, along with the point-by-point response, addresses the raised concerns.

The derivation of equations (albeit simple) is at some points not clear, therefore it is not possible to judge their correctness (see comments in the PDF).

The comments in the PDF mostly question the approximations required to derive equation (11) from Darcy's law:

$$q_n = (P\, k_a)\mathbf{grad}(P) = (k_a)\mathbf{grad}\left(\frac{P^2}{2}\right)$$

There is no approximation involved in this equation as **grad**(P^2/2) = P.**grad** (P) is a formula resulting from the mathematical properties of the derivative. If needed, we may add reference in the manuscript to a textbook deriving this equation.

The conceptual model (Fig. 10) needs some reformulation and a clearer explanation. As such it is not very convincing, although it is hard to judge if the reason is only poor text or not well elaborated concept.

As the focus of the paper is on the permeability of the limestone massif, we did not anticipate that providing details on the pressurization process was useful. As pointed out by both referees, the pressurization of the cave is an intriguing phenomenon that requires some explanation. To clarify this point, additional information is provided to show that pressurization events occur when there are waves on the cliff and are not correlated to rain infiltration. To support our assumption of a causal relationship between high waves and pressurization events, the significant height of the waves in front of the cave (data provided by the French Naval Hydrographic and Oceanographic Service, SHOM) and the daily rainfall will be added to Figure 4 in the manuscript (Figure 1 in this reply). While storms are often associated with rainfall, events with high rainfall but no high waves occasionally occur (e.g. August 2018) and they do not cause pressure variations in the cave. On the other hand, some pressurization events are not associated with heavy rainfall, but they are systematically associated with waves. Ongoing statistical analysis suggests a wave height threshold of about 1 m, but this threshold also depends on other factors (sea level, wave direction…). This will hopefully be the subject of a separate publication.

As pointed out by both referees, we will add at the beginning of the paper, in section 2 (after line 99) more explanation about the several paths (or levels) of connection between the sea and the cave through the limestone massif. We will redraw Figure 2B to show that the upper pathway is below the sea-level at a deepest level (so not "placed at the sea-level" as reported by referee 1), and redraw in Figure 10 (conceptual model) the upper pathway at a shallow level (still below the sea-level of course). Given the existence of karst pathways at several levels at the beginning of the paper will help the reader to conceptualize that waves can generate bubbles and then force seawater and air to flow through the

limestone massif. The detailed mechanism of how bubbles of air can flow inside the submerged karst is out of the scope of this paper.

**Line by line responses:**

l.41-42: This depends on the shape of the cave; downsloping cave would be active in winter, and vice versa.:

Modified sentence: "These flows are subject to seasonality, with generally stronger flows in winter and stratification of air masses in summer *for descending conduits and conversely for ascending conduits*".

l.63-64: What kind of events ?

As pressurization events have not been addressed at this point, "events" is replaced by periods

l.93-94: I guess you mean intergranular porosity ?

Modified: "Neither macro nor micro porosity has been observed" (Lamarche et al., 2012).

l.101-103: It would be helpful to add another perspective of the cave. I guess this is a DMR of ground; why don't you add a side perspective to get a feeling of the volume. I guess that the scale on Figure 2a does not apply for Figure 2b.

You are right, scale only applies to Fig. 2A. Figure 2B is not based on a 3D model, it's a drawing showing a cross-section of the Cosquer cave in its environment (sea, cliff). We believe Fig2B is more useful to understand the case study than a cross-section of the main rooms with archeological artwork (the totality of the cave is not yet available in 3D). We will had a vertical scale to Fig2B or a few elevation references to Figure 2B.

l.125: do you mean transmissivity and permeability of rock with respect to the air.

Absolutely

l.125: ms^-2:

Corrected

l.133-135: Since you are mentioning the phenomena and previous works, maybe mention what is the idea of mechanism behind it...:

Updated: "Data show that air pressure in the Cosquer cave is always higher than outside atmospheric pressure (Fig. 4). This very peculiar feature had already been shown by previous works (Vouvé et al., 1996; Arfib et al., 2018) and has now been confirmed on the timescale of several years of continuous survey (2014-2020). Air pressure in the cave and water level of the pools are correlated. *When the air pressure increases, the air is confined by the walls of the cave and pushes down the water table to balance the overpressure.* Conversely, between late spring…"

l.157: These events are highly interesting. I guess you should tell something about their origin already at this point.

As already mentioned in a previous comment, we will add more explanations about the mechanism that force the air to enter inside the cave: "Cave air pressure increases by the inflow of outside air, during periods with high waves on the cliff. Waves can produce and force air bubbles to propagate by submarine open fissures or karst conduits." We will also update the Figure 4 with wave height and daily rainfall, to show the causal relationship between high waves and pressurization events. Furthermore, the absence of significant wave activity during summer months supports the observation of rising water levels during this period. This statement will be added to the section.

l.216: Cumulative ?

Corrected

l.242-243: Here you probably mean air filled volume.

Absolutely. We tried to define the volume of the cave l.226-227, but it may still unclear. "Air-filled volume" is a welcome clarification.

Updated line 226: "$V$ the air-filled volume of the cave, defined as the volume of all the connected voids above the water level,"
Updated line 242: "Nevertheless, the variation of the air-filled volume of the cave due to tidal variations…"

l.263: if you write this in form 1/(1-palTah/pahTaL), equation 6 and 7 would be more obvious....

Corrected

l.282: Where do you get Qn from. Is it from dn(t)/dt from Eq. 9 ?

Absolutely yes, the equation will be added to the manuscript.

l.290: Explain P^2 in the right-hand side. It looks like that P from the left equation goes into the argument of grad.

It is a fundamental property of the gradient operator: $\mathrm{grad}(xy) = x\,\mathrm{grad}(y) + y\,\mathrm{grad}(x)$ or in the case $x = y$ : $\mathrm{grad}(x^2) = 2x\,\mathrm{grad}(x)$ (Lang, 1999).

Formulation of Eq. 11 is also given by Lefebvre (2003) and Charbeneau (2006).

l.295: I don't get the meaning of "defining the percolation threshold".

This sentence was incorrect (incomplete), sorry. The last part has been deleted.

l.303: I cannot get the sense from this sentence. Please reformulate.

Corrected: "[…] it follows from Eq. (11) that the total flux depends linearly on the difference of the squared pressure between the boundary conditions."

l.306: Refer to my comment to Eq. 11.... I am not sure what approximations were used to derive this Equation...

To pass from Eq. 11 to Eq. 13, no approximation is required.

l.309-310: This sentence become clear only after one reads the next paragraphs and Fig. 8. Please reformulate... Initially tell clearly that you assume three different geometries shown on Fig. 8., so that the reader is not confused:

Reformulation: "The interpretation of the air effective transmissivity coefficient $\lambda_a$, which has dimension of $m^3$, can vary based on the geometrical configuration. Three geometries are considered: a porous rock volume (Fig. 8A), a single fracture (Fig. 8B) and a pipe (Fig. 8C)."

l.476-481: You have not directly shown what causes the pressurization events. Nevertheless, I miss this clear statements in the introductory part. What I can get is that you have multiple pressurization events in the cold season with relaxation of this overpressure through the summer season. Could you relate pressurization events to some other outside atmospheric or oceanographic data (observation of winds/waves).

See below.

l.489-491: see my previous comment... can you show some correlation between pressurization and external events.

See below.

l.502-506: Although this is somehow central for the manuscript, it is hard to understand what you want to say. How is the air "pushed" by the waves and why does this happen only in the upper conduit and not in the lower one. Event though this is said to be outside the scope of the paper, one would still need some concept that relates the waves with air inflow.

It must be clarified that both the upper karst and lower karst conduit ends in sumps. With a depth of -37 m, the lower conduit (the human entrance) is too deep to be the air intake point. The sump of the upper conduit is short and shallow as showed in Figure 2B and the water level in the pool on the cave side varies with waves outside. The conduits connecting this pool to the sea have not been fully surveyed yet, but it is very likely that large waves can push air through them. As previously replied in the first comments, we will introduce the mechanism that is supposed to force the air inside the cave at the beginning of the paper (a few words in the introduction, then in section 2 (case study) and then in section 3 (pressurization events)), so that the conceptual model presented in section 6.2 can be clearly understood. We will also slightly modify the figures 2B and 10.

[Figure]

*Figure 1: Pressure, water level and temperature time series recorded in the Cosquer cave and at the Port-Miou observatory for years 2017 and 2018: (A) Sea level at Port-Miou ($h_s$) and Cosquer cave water level ($h_w$), expressed in column of seawater ($m_{sw}$) above the probe with the same reference level. The green dashed line shows the bottom of the horses panel (paleolithic*

*decorated wall). (B) Atmospheric pressure (Patm) outside the cave and cave air pressure (Pa). (C) cave air temperature (Ta) and cave water temperature in Room 1 (Tw). (D) daily rainfall and (E) waves significant height in front of the Cosquer cave. Pressurization events periods are highlighted in grey.*

**Bibliography**

Charbeneau, R.J., 2006. Groundwater Hydraulics and Pollutant Transport. Waveland Press.

Lamarche, J., Lavenu, A.P.C., Gauthier, B.D.M., Guglielmi, Y., Jayet, O., 2012. Relationships between fracture patterns, geodynamics and mechanical stratigraphy in Carbonates (South-East Basin, France). Tectonophysics 581, 231–245. https://doi.org/10.1016/j.tecto.2012.06.042

Lang, S., 1999. Fundamentals of differential geometry, Graduate texts in mathematics. Springer, New York Berlin Heidelberg.

Lefebvre, R., 2003. Écoulement multiphase en milieux poreux. Université Laval/INRS-Eau, Terre et Environnement.

---

## Author Response (AR1)

**Author's response to editor (May 29, 2024)**

**Mesoscale permeability variations estimated from natural airflows in the decorated Cosquer Cave (SE France)**

EGUSPHERE-2023-2380

Hugo Pellet [1,2], Bruno Arfib [1], Pierre Henry [1], Stéphanie Touron [2], Ghislain Gassier [1]

[1] Aix Marseille Univ, CNRS, IRD, INRAE, CEREGE, Aix-en-Provence, France

[2] UAR3224-CRC-Laboratoire de Recherche des Monuments Historiques, France

*Correspondence to*: Hugo Pellet (pellet@cerege.fr)

Dear Pr. Gerrit de Rooij,

We appreciate the invitation to revise our manuscript based on the reviewers' comments.

Both reviewers highlighted the lack of explanation for the origin of the pressurization events despite the originality of the phenomenon. This, in turn, led to confusion regarding the presented conceptual model. To address the reviewers' comments, we applied the corrections we had proposed in response to the reviewers at the end of the *HESS open discussion*.

Revisions made on the manuscript are explained in the following pages, split in two sections:

1. Major changes: This section details the significant revisions made to address the pressurization events, conceptual model, and equations.
2. Line-by-line responses to the reviewers' comments and other improvements.

We believe these revisions effectively address the reviewers' concerns and significantly strengthen the manuscript. Following the reviewer's advice, the manuscript was corrected by a native English-speaker. Reviewers' comments are written in *italic*.

Changes are referenced by line number of the reviewed manuscript.

Yours sincerely,

Hugo Pellet

**1. Major changes**

**1.1. Pressurization events and conceptual model**

Both reviewers flagged a lack of clarity in our conceptual model of the Cosquer cave, particularly regarding the mechanism behind the pressurization events and air inflows. To address these concerns and enhance clarity, we made the following revisions to the paper:

**l.63-66**: we describe and stated the supposed mechanism in the introduction section:

"Cave air pressure increases by the inflow of outside air, during periods with high waves breaking on the cliff of the coastal limestone massif. Waves can produce and force air bubbles to flow along submarine open fissures or karst conduits inside the massif during short periods of time. Since the rock is not airtight, air slowly flows out through the limestone massif over several months."

**l.100-105**: A paragraph has been added detailing the connection between the cave and the sea through the shallow sumps:

"This pool is connected to the sea outside by karstic conduits that have not been fully surveyed for accessibility and safety reasons. However, the water level in the pool on the cave side varies with waves outside, which indicates that communication with the sea occurs at a shallow depth (Figure 2), and it is suspected that large waves can push air bubbles through the conduits. Expeditions in the cave are never scheduled during rough weather, which makes direct observation of this phenomenon nearly impossible. However, we will show that instrumental records display a strong correlation between episodes of high waves and air input into the cave."

**l.126-128**: waves data are presented:

"The significant height of waves in front of the cave is the result of simulations provided by the French Naval Hydrographic and Oceanographic Service (SHOM, 2024)."

**l.141**: the subsection *3.1 Seasonal pressure variations* is split into 2 subsections: 3.1 Overpressure in the Cosquer cave and 3.2 Seasonal variations.

**l.141-164** (**3.1 Overpressure in the Cosquer cave**):

In this subsection, the mechanism behind the correlation between cave air pressure and water level is now clearly defined (**l.144-146**):

"Air pressure in the cave and water level of the pools are anti-correlated (Fig. 4A and Fig. 4B). When the air pressure increases, the air is confined by the walls of the cave and pushes down the water table to balance the overpressure."

We made a qualitative description of seasonal cave air pressure variations with emphasize on the pressurization events (**l.150-154**):

"A succession of pressure peaks occurs between October and May (highlighted in grey in Fig. 4) and these are generally absent in summer. These sharp rises in air pressure over tens of minutes to a few hours followed by a rapid pressure decay (over a day or so) are referred to in this paper as pressurization events and the shape of these events will be described in more detail in the following section. Between 20/10/2017 and 30/04/2018, about 30 of these pressurization events occurred. These events generate the cave overpressure by the inflow of outside air."

We now compare the pressurization events periods with daily rainfall (Fig. 4D) and waves significant height (Fig. 4E). This comparison makes it possible to reject the hypothesis of a pressure increase caused by air flushed into the fractures of the unsaturated zone during recharge events and shows a strong relationship with waves height in front of the cave (**l.154-159**):

"They occur systematically during periods with high waves in front of the coastal limestone massif (Fig. 4E). While storms are often associated with rainfall, periods with high rainfall but no high waves occasionally occur (e.g. August 2018, Fig. 4D) and they do not cause pressure variations in the cave (Fig. 4B). On the other hand, some pressurization events are not associated with heavy rainfall, but they

are systematically associated with waves (Fig. 4E). Furthermore, the absence of significant wave activity during summer months supports the observation of rising water levels during this period."

We assume that air flows into the cave through shallow fractures and conduits and we propose a mechanism for air inflows through wave-generated bubbles (**l.159-164**):

"The current hypothesis is that breaking waves and waves crashing against the cliff can generate bubbles and then force seawater and air to flow through the limestone massif by shallow fissures and karst conduits (however, the detailed mechanism of how bubbles of air can flow inside submerged karst is outside the scope of this paper). Given the existence of karst pathways at several levels connecting the sea and the cave, air flows through shallow sumps and reaches the cave by upper conduits away from the decorated rooms, and not the lower conduit (human entrance) which is too deep to be the air intake point (-37 m)."

**l.165-178** (**3.2 Seasonal variations**):

Quantitative description of seasonal variation of cave air pressure, water level in the cave and cave air and water temperature is made:

"As previously described, air overpressure in the cave decreases during summer. The summer depressurization rate was in average -0.21 $cm_{sw}$ day$^{-1}$ in 2017 and -0.32 $cm_{sw}$ day$^{-1}$ in 2018 (mean over July and August). Conversely, during pressurization events, net air pressure usually increases in the cave, i.e. the air pressure is usually higher after the event than before. Two thresholds are graphically identified in Fig. 4B: (i) maximum air pressure never exceeds 11.5 $m_{sw}$ (1.16 hPa); (ii) immediately after pressurization peaks, the air pressure drops down to an overpressure level between 10.8 $m_{sw}$ (1.09 hPa) and 10.7 $m_{sw}$ (1.08 hPa). Below this level, the pressure decrease rate slows down considerably. The lowest water level is about 1.5 $m_{sw}$ below the seawater level (0.40 $m_{sw}$ above the probe, Fig. 4A) during winter. Thus, at a seasonal time scale, the pressure variation range (Fig. 4A) is around 1.5 $m_{sw}$ (0.15 hPa). […]"

**In section 6.2 Air renewal**:

**l.522**: added "shallow submarine karst inception horizons and fractures" for clarity.

**1.2. Model and equations**

A major concern for both reviewers was about the equations used in the section 4:

**l.294**: *if you write this in form 1/(1-palTah/pahTaL), equation 6 and 7 would be more obvious:* Eq. (5) modified to match the expected form: $V_l = S_w \, \Delta h_w \, \dfrac{1}{1 - \frac{P_{al} T_{ah}}{P_{ah} T_{al}}}$

**l.314**: *Where do you get Qn from. Is it from dn(t)/dt from Eq. 9 ?:*

equation added: "[…] $Q_n = \dfrac{dn(t)}{dt}$ is the molar flow rate (mol s$^{-1}$) […]"

**l.322**: Both reviewers asked about the factor 2 and the P$^2$ in Eq. (11)*:*

no modification was required in this equation but references (Lang, 1999 and Charbeneau, 2006) are added to support our statement.

**l.338**: *Refer to my comment to Eq. 11... I am not sure what approximations were used to derive this Equation:*

to pass from Eq. 11 to Eq. 13, no approximation is required.

**l.338***: what drives the temperature variations in the cave air? Using a constant Trock is an approximation as we know there are seasonal fluctuations in both, the cave air and the outside atmosphere? what is their effect:*

First, we added the water temperature in the cave to Fig. 4C to show that water temperature in the cave drives the air temperature in the cave (l.174-178):

"Air temperature varies in the range 16°C to 21°C, in a seasonal pattern. The maximum is observed at the end of summer and the minimum at the beginning of spring (Fig. 4C). The air temperature variations are mainly driven by the water temperature variations. Water temperature variations are related to exchange with seawater outside the cave through conduits. Air temperature variations are smoother than water temperature variations, and delayed because of heat exchange with the cave walls, which have thermal inertia."

In the authors' answer to reviewer 1, we made the demonstration that temperature variation is negligeable in the result, but we doubt it is necessary to assess it in the manuscript as the corresponding uncertainty is much smaller that the uncertainty on the air flux measurement.

**2. Line by line responses**

**[1]** Following the reviewer's advice, the manuscript was corrected by a native English-speaker. These corrections are not detailed below but are visible in the track-change file.

**Reviewer 1**

**[2] l.37**: "2" replaced by "two".

**[3] l.42**: *underline that the stronger airflow in winter is valid for a descending conduit. In ascending conduits, this would likely happen in summer*:

Modified sentence: "These flows are subject to seasonality, with generally stronger flows in winter and stratification of air masses in summer for descending conduits and conversely for ascending conduits".

**[4] l.63-65:** *please state the supposed mechanism driving this airflow. Obviously, this wouldn't happen without an external force if the cave is over-pressured*:

the supposed mechanism is stated: "Cave air pressure increases by the inflow of outside air, during periods with high waves breaking on the cliff of the coastal limestone massif. Waves can produce and force air bubbles to flow along submarine open fissures or karst conduits inside the massif during short periods of time."

**[5] l.58**: added: "(southeastern France)".

**[6] l.81**: space between numbers removed: "32500" and "19000".

**[7] l.86**: corrected sentence: "The sea was lower back then".

**[8] l.86-87**: space between numbers removed: "20000" and "8000".

**[9] l.91**: *Cretaceous should be capitalized*: corrected.

**[10] l.92**: *"thin sections observations" delete observations*: deleted.

**[11] l.123-125**: *However, no information is given about how the elevation of the water table inside the cave was determined. Is there an altimetric survey available?:*

added: "[…] pressure sensors of air inside and outside the cave are intercalibrated. When the same pressure is recorded inside and outside the cave, the water level in the cave is thus equal to the sea level.".

**[12] l.144**: *"monitoring" instead of "survey"*: "survey" replaced by "monitoring".

**[13] l.174-178:** *what is the source of the temperature fluctuations measured in the cave air? Does it correlate to the seawater temperature and are these temperature fluctuations compensated in the pressure analysis ?:*

water temperature is added to Fig. 4C and temperature variations are described l.174-178 : "Air temperature varies in the range 16°C to 21°C, in a seasonal pattern. The maximum is observed at the end of summer and the minimum at the beginning of spring (Fig. 4C). The air temperature variations are mainly driven by the water temperature variations. Water temperature variations are related to exchange with seawater outside the cave through conduits. Air temperature variations are smoother than water temperature variations, and delayed because of heat exchange with the cave walls, which have thermal inertia."

**[14] l.218**: *tide-related temperature variations "in the cave air" (?)*:

Corrected: "Tide-related temperature variations in the cave air are observed".

**[15] l.222-l.224**: *phrasing: According the cave air temperature measurements it is rather the wall which stays close to equilibrium with the cave air rather than the opposite*:

modified sentence: "These observations indicate that thermal convection is very active at least in the decorated rooms of the cave and that at the tide time scale, the air in the cave and the walls remains close to thermal equilibrium."

**[16] l.225**: *Fig 6 suggests that, during low tide, the water level is higher in the cave than at Port Miou. Can you expand on this?*

Water levels are centered by their mean, see l.205-206: [...] pressures expressed in meter of seawater (msw) and mean-centered at a two days scale [...]

**[17] l.225**: *In Fig 6, you show that the cave water level is c. 5 cm lower than the sea level whereas the fluctuations shown on Fig 7 range in the order of 1 m. What do I get wrong?*

Fig. 6 compares tide-related water level and sea level variation (around few centimeters) whereas Fig. 7 presents the annual amplitude of cave water level (up to 1.5 msw). See the next comment for clarifications.

**[18] l.232-234**: *you may want to explain better why the sea-level was c. 0.25 cm higher in 2018 than 2017. What do the horizontal lines (red and black) represent on the figure?*:

sentence added : "The highest level recorded in 2017 (water level close to 1,8 meters above the probe in Fig. 7) was an exceptionally high water level in the cave, equivalent to the sea-level for a few days. A scaled photo of the Horses panel is added to the figure. It illustrates how high the water level can rise and flood the artwork.". Horizontal lines on Fig. 7 were confusing and are thus removed.

**[19] l.271**: *This variation is :* corrected.

**[20] l.322**: *where does the factor 2 on the right hand-side come from?*: See section 1.2 of this document.

**[21] l.335**: *what drives the temperature variations in the cave air? Using a constant Trock is an approximation as we know there are seasonal fluctuations in both, the cave air and the outside atmosphere? what is their effect?* See comment [13] for the source of temperature variations in the cave air. Water temperature in the cave is added to the Fig. 4C.

**[22] l.339**: added "much"

**[23] l.358**: *please edit: 'equation will, in most cases, yields ...'*:

Corrected: "In most cases, eq. (15) yields a correct order [...]"

**[24] l.379-386**: *unclear please edit*:

paragraph rephrased to improve clarity: "[...] Using this latter value, the calculated air-filled volume of the cave was the same in 2017 and 2018, within the range of uncertainties, as expected. It shows that using data of the two summer months, volume can be calculated by the method proposed. Table 2 summarizes mean cave water level measurement and volume calculated over the two summer months for years 2015 to 2020. Mean summer cave volume over the 6 years was 5000 m³ for an average water level of 1.54 cm$_{sw}$. This mean volume was maximal in 2020 when the mean water level was minimal,

and was minimal in 2016 when the mean water level was maximal. Using the complete dataset available from years 2015 to 2020, the 6-year average water level was 1.33 $m_{sw}$ and yields an average air-filled volume of the cave of 5184 $m^3$."

**[25] l.415**: *missing article:*

corrected: "[…] when the air […]"

**[26] l.438-442**: *unclear, please rephrase: are your referring to the rapid pressure decay or the slow depressurization:*

rephrased: "[…] Similarly, the cumulative net annual air outflow during rapid pressure decays varied from 7720 $m^3$ (year 2015) to 18260 $m^3$ (year 2020) with an annual average of 13270 $m^3$. The budget gives a remaining outflowing air volume (6-year average, 4320 $m^3$) corresponding to the cumulative net annual air outflow during slow depressurization periods."

**[27] l.471**: *have been*: corrected

**[28] l.486**: *you may want to specify that this is during summer*: corrected: "Over the two summer months, there is a slow decrease […]"

**[29] l.488**: *the pools' reference surface (?)*: Corrected

**[30] l.491**: *should be "and during years"*: Corrected

**[31] l.490-491**: *waves where not addressed yet, did they? Wouldn't it rather be associated with groundwater recharge?*

Waves and rainfall are now presented earlier. We showed the correlation between pressurization events and waves height. see Section 1.1 of this document.

**[32] l.490**: *Wave heights and direction were not discussed so far*. See previous comment

**[33] l.538**: *This upper conduit was not introduced yet*.

A paragraph has been added l.100-103, detailing the link between the cave and the shallow sumps. We postulate that air passes through these sumps during high waves periods. See section 1.1 of this document for the details.

**[34] l.587**: *not connected to the cave. But what about the epikarst, flushing air into the fracture during recharge events?*

See section 1.1 of this document

**Reviewer 2**

**[35] l.42**: *This depends on the shape of the cave; downsloping cave would be active in winter, and vice versa:*

corrected: "[…] for descending conduits and conversely for ascending conduits […]".

**[36] l.63-65**: *What kind of events ?*:

Pressurization events and the supposed mechanism are presented. See section 1.1 of this document.

**[37] l.94**: *I guess you mean intergranular porosity?:*

modified: "[…] carbonates do not display neither macro nor micro porosity."

**[38] l.110**: *It would be helpful to add another perspective of the cave. I guess this is a DMR of ground; why don't you add a side perspective to get a feeling of the volume. I guess that the scale on Figure 2a does not apply for Figure 2b.*

Figure 2B is not based on a 3D model, it's a drawing showing a cross-section of the Cosquer cave in its environment (sea, cliff). We believe Fig2B is more useful to understand the case study than a crosssection of the main rooms with archeological artwork (the totality of the cave is not yet available in 3D). See below for Figures and tables changes.

**[39] l.135**: *do you mean transmissivity and permeability of rock with respect to the air:*

"Air intrinsic transmissivity" changed to "Intrinsic transmissivity to air"

"Air intrinsic permeability" changed to "Intrinsic permeability to air"

**[40] l.144-147**: *Since you are mentioning the phenomena and previous works, maybe mention what is the idea of mechanism behind it...:*

Paragraph modified: "Air pressure in the cave and water level of the pools are anti-correlated (Fig. 4A and Fig. 4B). When the air pressure increases, the air is confined by the walls of the cave and pushes down the water table to balance the overpressure. Conversely, between late spring and early autumn, there is a slow decrease in cave air pressure and the water level simultaneously increases."

**[41] l.157**: *These events are highly interesting. I guess you should tell something about their origin already at this point:*

See section 1.1 of this document.

**[42] l.231**: *Cumulative ?:*

"cumulated" replaced by "cumulative" throughout the manuscript.

**[43] l.257**: *Here you probably mean air filled volume:*

Corrected throughout the manuscript

**[44] l.294**: *if you write this in form 1/(1-palTah/pahTaL), equation 6 and 7 would be more obvious.*

Corrected, see section 1.2 of this document.

**[45] l.313**: *Where do you get Qn from. Is it from dn(t)/dt from Eq. 9 ?*

See section 1.2 of this document.

**[46] l.322**: *Explain P^2 in the right-hand side. It looks like that P from the left equation goes into the argument of grad.*

See section 1.2 of this document.

**[47] l.326**: *I don't get the meaning of "defining the percolation threshold":*

this sentence was incomplete and confusing. The last part has been deleted "[…] low water saturation to host a continuous gas phase."

**[48] l.335**: *I cannot get the sense from this sentence. Please reformulate*:

Sentence modified: "[…] it follows from Eq. (11) that the total flux depends linearly on the difference of the squared pressure between the boundary conditions."

**[49] l.338**: *Refer to my comment to Eq. 11.... I am not sure what approximations were used to derive this Equation.*:

See section 1.2 of this document.

**[50] l.341-343**: […] *Initially tell clearly that you assume three different geometries shown on Fig. 8., so that the reader is not confused*: modified: "The interpretation of the air effective transmissivity coefficient $\lambda_a$, which has dimension of m$^3$, can vary based on the geometrical configuration. Three geometries are considered: a porous rock volume (Fig. 8A), a single fracture (Fig. 8B) and a pipe (Fig. 8C)."

**[51] l.512-517**: *You have not directly shown what causes the pressurization events. Nevertheless, I miss this clear statements in the introductory part. What I can get is that you have multiple pressurization events in the cold season with relaxation of this overpressure through the summer season. Could you relate pressurization events to some other outside atmospheric or oceanographic data (observation of winds/waves).*:

See section 1.1 of this document.

**[52] l.525-523**: *see my previous comment... can you show some correlation between pressurization and external events*:

See section 1.1 of this document.

**[53] l.538-543**: *Although this is somehow central for the manuscript, it is hard to understand what you want to say. How is the air "pushed" by the waves and why does this happen only in the upper conduit and not in the lower one. Events though this is said to be outside the scope of the paper, one would still need some concept that relates the waves with air inflow.*

See section 1.1 of this document.

**Tables and figures**

**[54] Table 1**: reviewer 1 asked to add to Table 1 a column for numerical estimates and a column for references. Among the 35 variables presented in the Table 1, only 9 of them are constant values ($P_0$, $T_0$, $S_w$, $g$, $R$, $\gamma$, $\rho_{sea}$, $h_0$, $\mu$), most of the new columns would thus be empty, we therefore preferred to not add them. However, all the model input values are presented in the text.

**[55] Table 1**: gravitational acceleration unit corrected

**[56] Fig. 2A**: modifications are provided to improve the readability of the map.

**[57] Fig. 2B**: the upper sump is redrawn to emphasize that it is submerged. An approximative scale and approximative elevation are added.

**[58] Fig. 2B**: *It would be helpful to add another perspective of the cave*:

we do not have a better cross-section of the cave yet.

**[59] Fig. 4**: Daily rainfall (Fig. 4D) and waves significant height in front of the Cosquer cave are added to support our assumptions and conceptual model.

**[60] Fig. 7**: Horizontal lines are removed for better clarity.

**[61] Fig. 10**: modified to better match the statements and interpretations made along the paper.

---

## Referee Report (RR1)

[referee-annotated manuscript omitted]

---

## Author Response (AR2)

**Author's response to editor (July 8, 2024)**

**Mesoscale permeability variations estimated from natural airflows in the decorated Cosquer Cave (SE France)**

EGUSPHERE-2023-2380

Hugo Pellet [1, 2], Bruno Arfib [1], Pierre Henry [1], Stéphanie Touron [2], Ghislain Gassier [1]

[1] Aix Marseille Univ, CNRS, IRD, INRAE, CEREGE, Aix-en-Provence, France

[2] UAR3224-CRC-Laboratoire de Recherche des Monuments Historiques, France

*Correspondence to*: Hugo Pellet (pellet@cerege.fr), Bruno Arfib (arfib@cerege.fr)

Dear Pr. Gerrit de Rooij,

We would like to express our sincere appreciation to the editor for the time you dedicated to review our article.

Please, find below our responses to the editor's comments (*in italic*). Comments with their answers are listed below and referenced by line number of the reviewed manuscript.

Yours sincerely,

Hugo Pellet

**[1] l.13** *Perhaps briefly explain this is caused by bubbles carried by the sea water into the cave?*

Short explanations are added : "Although the cave air is confined by the rock and the seawater, there are also external air inflows during short pressurization events, *in connection with waves that can produce and force air bubbles to flow along submarine open fissures or karst conduits inside the massif*."

**[2] l.147** : *Replace "anti-correlated" by "negatively correlated"*

Corrected

**[3] l.159** : *Why 'On the other hand'? This suggests conflictng observations, but they are in agreement: waves, not rain, occur whenever pressurization occurs. I suggest "Thus, some pressurization..."*

Corrected

**[4] l.160** : *This is consistent with the observed rising water levels during the summer months, when there is no significant wave activity.*

Corrected

**[5] l.162** : *Well, not really, is it? But I agree it would be too much to ask you to go out and observe it. You cannot send divers out under these conditions, and to install some kind of underwater camera system to film bubbles just for the sake of this paper is not realistic. I think you can safely delete this sentence without running the risk that you will be asked to back up your hypothesis with direct observations. Bubbles have higher than atmospheric pressure by definition - they are under water after all. Wave activity causes more air to dissolve in the water, increasing the likelihood of bubble formation and slowing down their dissolution in water. All in all there is quite a bit of physics to back up your hypothesis, even without direct observations.*

Thanks for the suggestion, the sentence was deleted.

**[6] l.254** : *In summer, the sea water temperature rises, therefore the solubility of nitrogen and oxygen gas will increase. Could it be that some dissolution of the two main components of air in the warming sea water also contributes to the depressurization in summer?*

Solubility of air nitrogen and oxygen actually decrease with the increase in water temperature (Battino et al., 1984; Fernández-Prini et al., 2003). This phenomenon has not been considered.